# Value of uncertain streamflow observations for hydrological modelling

Simon Etter[1], Barbara Strobl[1], Jan Seibert[1,2], Ilja van Meerveld[1]

[1]Department of Geography, University of Zurich, Winterthurerstrasse 190, 8057 Zurich, Switzerland
[2]Department of Earth Sciences, Uppsala University, Villavägen 16, 752 36 Uppsala, Sweden

*Correspondence to*: Simon Etter (simon.etter@geo.uzh.ch)

**Abstract.** Previous studies have shown that hydrological models can be parameterized using a limited number of streamflow measurements. Citizen science projects can collect such data for otherwise ungauged catchments but an important question is whether these observations are informative given that these streamflow estimates will be uncertain. We address the value of inaccurate streamflow estimates for calibration of a simple bucket-type runoff model for six Swiss catchments. We pretended that only a few observations were available and that these were affected by different levels of inaccuracy. The level of inaccuracy was based on a log-normal error distribution that was fitted to streamflow estimates of 136 citizens for medium-sized streams. Two additional levels of inaccuracy, for which the standard deviation of the error-distribution was divided by two and four, were used as well. Based on these error distributions, random errors were added to the measured hourly streamflow data. New time series with different temporal resolutions were created from these synthetic streamflow time series. These included scenarios with one observation each week or month, as well as scenarios that are more realistic for crowdsourced data that generally have an irregular distribution of data points throughout the year, or focus on a particular season. The model was then calibrated for the six catchments using the synthetic time series for a dry, an average and a wet year. The performance of the calibrated models was evaluated based on the measured hourly streamflow time series. The results indicate that streamflow estimates from untrained citizens are not informative for model calibration. However, if the errors can be reduced, the estimates are informative and useful for model calibration. As expected, the model performance increased when the number of observations used for calibration increased. The model performance was also better when the observations were more evenly distributed throughout the year. This study indicates that uncertain streamflow estimates can be useful for model calibration but that the estimates by citizen scientists need to be improved by training or more advanced data filtering before they are useful for model calibration.

## 1   Introduction

The application of hydrological models usually requires several years of precipitation, temperature and streamflow data for calibration, but these data are only available for a limited number of catchments. Therefore, several studies have addressed the question: how much data are needed to calibrate a model for a catchment? Yapo et al. (1996) and Vrugt et al. (2006a)

using stable parameters as a criteria for satisfying model performance, concluded that most of the information to calibrate a model is contained in 2-3 years of continuous streamflow data and that no more value is added when using more than eight years of data. Perrin et al. (2007) using the Nash-Sutcliffe efficiency criterion (NSE), showed that streamflow data for 350 randomly sampled days out of a 39 year period were sufficient to obtain robust model parameter values for two bucket-type

models, TOPMO, which is derived from TOPMODEL concepts (Michel et al. 2003), and GR4J (Perrin et al., 2003). Brath et al. (2004) using the volume-, relative peak- and time to peak error concluded that at least three months of continuous data were required to obtain a reliable calibration. Other studies have shown that discontinuous streamflow data can be informative for constraining model parameters (Juston et al., 2009; Pool et al., 2017; Seibert and Beven, 2009; Seibert and McDonnell, 2015). Juston et al. (2009) used a multi-objective calibration that included groundwater data and concluded that

the information content of a subset of 53 days of streamflow data was the same as 1065 days of data from which the subset was drawn. Seibert and Beven (2009) using the NSE criterion, found that model performance reached a plateau for 8-16 streamflow measurements collected throughout a one-year period. They, furthermore, showed that the use of streamflow data for one event and the corresponding recession resulted in a similar calibration performance as data for the six highest measured streamflow values during a two-month period.

These studies had different foci and used different model performance metrics but nevertheless their results are encouraging for the calibration of hydrological models for ungauged basins based on a limited number of high-quality measurements. However, the question remains: how informative are low(er)-quality data? An alternative approach to high quality streamflow measurements in ungauged catchments is to use citizen science. Citizen science has been proven to be a valuable tool to collect (Dickinson et al., 2010) or analyse (Koch and Stisen, 2017) various kinds of environmental data, including

hydrological data (Buytaert et al., 2014). Citizen science approaches use simple methods to enable a large number of citizens to collect data and allow local communities to contribute data to support science and environmental management. Citizen science approaches can be particularly useful in light of the declining stream gauging networks (Ruhi et al., 2018; Shiklomanov et al., 2002) and to complement the existing monitoring networks. However, citizen science projects that collect streamflow or stream level data in flowing water bodies are still rare. Examples are the CrowdHydrology project

(Lowry and Fienen, 2013), SmartPhones4Water in Nepal (Davids et al., 2018) and a project in Kenya (Weeser et al., 2018), which all ask citizens to read stream levels at staff gauges and to send these via an app or as text message to a central database. Estimating streamflow is obviously more challenging than reading levels from a staff gauge but citizens can apply the stick- or float method, where they measure the time it takes for a floating object (e.g. a small stick) to travel a given distance to estimate the flow velocity. Combined with estimates for the width and the average depth of the stream, this

allows them to obtain a rough estimate of the streamflow. However, these streamflow estimates may be so inaccurate that they are not useful for model calibration. It is therefore necessary to not only evaluate the requirements of hydrological models in terms of the amount and temporal resolution of data, but also in terms of the achievable quality by the citizen scientists before starting a citizen science project.

The effects of rating curve uncertainty on model calibration (e.g. McMillan et al. 2010; Horner et al. 2018) and the value of sparse datasets (Davids et al., 2017) have been quantified in recent studies. However, the potential value of sparse datasets in combination with large uncertainties (such as those from crowdsourced streamflow estimates) has not been evaluated so far. Therefore, the aim of this study was to determine the effects of observation inaccuracies on the calibration of bucket-type hydrological models when only a limited number of observations are available. The specific objectives of this paper are to determine (*i*) whether the streamflow estimates from citizen scientists are informative for model calibration or if these errors need to be reduced (e.g. through training) to become useful and (*ii*) how the timing of the streamflow observations affects the calibration of a hydrological model. The latter is important for citizen science projects, as it provides guidance on whether it is useful to encourage citizens to contribute streamflow observations during a specific time of the year.

## 2 Methods

To assess the potential value of crowdsourced streamflow estimates for hydrological model calibration, the HBV (Hydrologiska Byråns Vattenbalansavdelning) model (Bergström, 1976) was calibrated against streamflow time series for six Swiss catchments, as well as different subsets of the data that represent citizen science data in terms of errors and temporal resolution. Similar to the approach used in several recent studies (Ewen et al., 2008; Finger et al., 2015; Fitzner et al., 2013; Haberlandt and Sester, 2010; Seibert and Beven, 2009), we pretended that only a small subset of the data were available for model calibration. In addition, various degrees of inaccuracy were assumed. The value of these data for model calibration was then evaluated by comparing the model performance for these subsets of data to the performance of the model calibrated with the complete measured streamflow time series.

### 2.1 HBV model

The HBV model was originally developed at the Hydrologiska Byrans Vattenavdelning unit at the Swedish Meteorological and Hydrological Institute (SMHI) by Bergström (1976). The HBV model is a bucket-type model that represents snow, soil, groundwater and stream routing processes in separate routines. In this study, we used the version HBV-light (Seibert and Vis, 2012).

### 2.2 Catchments

The HBV-light model was set up for six 24-186 km$^2$ catchments in Switzerland (Table 1 and Figure 1). The catchments were selected based on the following criteria: *i*) there is little anthropogenic influence, *ii*) they are gauged at a single location, *iii*) they have reliable streamflow data during high flow and low flow conditions (i.e. no complete freezing during winter and a cross section that allows accurate streamflow measurement at low flows), and *iv*) there are no glaciers. The six selected catchments (Table 1) represent different streamflow regime types (Aschwanden and Weingartner, 1985). The snow dominated highest elevation catchments (Allenbach and Riale di Calneggia) have the largest seasonality in streamflow, i.e. the biggest differences between the long-term maximum and minimum Pardé coefficients, followed by the rain and snow

dominated Verzasca catchment. The rain dominated catchments (Murg, Guerbe and Mentue) have the lowest seasonal variability in streamflow (Table 1). The mean elevation of the catchments varies from 652 to 2003 m asl (Table 1). The elevation range of each individual catchment was divided in 100 m elevation bands for the simulations.

## 2.3    Measured data

Hourly runoff time series (based on 10 minute measurements) for the six study catchments were obtained from the Federal Office for the Environment (FOEN; see Table 1 for the gauging station numbers). The average hourly areal precipitation amounts were extracted for each study catchment from the gridded CombiPrecip dataset from MeteoSwiss (Sideris et al., 2014). This dataset combines gauge and radar precipitation measurements at an hourly timescale and 1 $km^2$ spatial resolution and is available since 2005.

We used hourly temperature data from the automatic monitoring network of MeteoSwiss (see Table 1 for the stations) and applied a gradient of -6 °C per 1000 m to adjust the temperature of each weather station to the mean elevation of the catchment. Within the HBV model, the temperature was then adjusted for the different elevation bands using a calibrated lapse rate.

As recommended by Oudin et al. (2005), potential evapotranspiration was calculated using the temperature-based potential

evapotranspiration model of McGuinness and Bordne, (1972) using the day of the year, the latitude and the temperature. This rather simplistic approach was considered sufficient because this study focused on differences in model performance relative to a benchmark calibration.

## 2.4    Selection of years for model calibration and validation

The model was calibrated for an average, a dry and a wet year to investigate the influence of wetness conditions and the

amount of streamflow on the calibration results. The years were selected based on the total streamflow during summer (July-September). The driest and the wettest years of the period 2006-2014 were selected based on the smallest and largest sum of streamflow during the summer. The average streamflow years were selected based on the proximity to the mean summer streamflow for all the years 1974-2014 (1990-2014 for Verzasca). For each catchment the years that were the 2nd-closest to the mean summer streamflow for all years, as well as the years with the 2nd-lowest and 2nd-highest streamflow sum were

chosen for model calibration (see

**Table 2). We did this separately for each catchment because for each catchment a different year was dry, average or wet. For the validation, we chose the year closest to the mean summer streamflow and the years with the lowest and the highest total summer streamflow (see**

Table 2). We used each of the parameter sets obtained from calibration for the dry, average or wet years to validate the

model for each of the three validation years, resulting in nine validation combinations for each catchment (and each dataset, as described below).

## 2.5 Transformation of data sets to resemble citizen science data quality

### 2.5.1 Errors in crowdsourced streamflow observations

Strobl et al. (in review) asked 517 participants to estimate streamflow based on the stick method at ten streams in Switzerland. Here we use the estimates for the medium sized streams Töss, Sihl and Schanzengraben in the Canton of Zurich

and the Magliasina in Ticino (n=136), which had a similar streamflow range at the time of the estimations (2.6 – 28 m³/s) as the mean annual streamflow of the six streams used for this study (1.2 – 10.8 m³/s). We calculated the streamflow from the estimated width, depth and flow velocities using a factor of 0.8 to adjust the surface flow velocity to the average velocity (Harrelson et al., 1994). The resulting streamflow estimates were normalized by dividing them by the measured streamflow. We then combined the normalized estimates of all four rivers and log transformed the relative estimates. A normal

distribution with a mean of 0.12 and a standard deviation of 1.30 fits the distribution of the log-transformed relative estimates well, with a standard error of the mean of 0.11 and a standard error of the standard deviation of 0.08 (Figure 2).

To create synthetic datasets with data quality characteristics that represent the observed crowdsourced streamflow estimates, we assumed that the errors in the streamflow estimates are uncorrelated (as they are likely provided by different people). For each time step, we randomly selected a relative error value from the lognormal distribution of the relative estimates (Figure

2) and multiplied the measured streamflow with this relative error. To simulate the effect of training and to obtain time series with different data quality, two additional streamflow time series were created using a standard deviation divided by two (standard deviation of 0.65) and by four (standard deviation of 0.33). This reduces the spread in the data (but did not change the small systematic overestimation of the streamflow), so that large outliers are still possible, but are less likely. To summarize, we tested the following four cases:

- *No error*: The data measured by the FOEN, assumed to be (almost) error-free, the benchmark in terms of quality.
- *Small error*: random errors according to the log-normal distribution of the snapshot campaigns with the standard deviation divided by 4.
- *Medium error*: random errors according to the log-normal of the surveys with the standard deviation divided by 2.
- *Large error*: typical errors of citizen scientists, i.e. random errors according to the log normal distribution of errors

from the surveys.

### 2.5.2 Filtering of extreme outliers

Usually some form of quality control is used before citizen science data are analyzed. Here, we used a very simple check to remove unrealistic outliers from the synthetic datasets. This check was based on the likely minimum and maximum streamflow for a given catchment area. We defined an upper limit of possible streamflow values as a function of catchment

area using the dataset of maximum streamflow from 1500 Swiss catchments provided by Scherrer AG, Hydrologie und Hochwasserschutz (2017). To account for the different precipitation intensities north and south of the Alps, different curves were created for the catchments on each side of the Alps. All streamflow observations, i.e., modified streamflow

measurements, above the maximum observed streamflow for a particular catchment size including a 20 % buffer (**Error! Reference source not found.**), were replaced by the value of the maximum streamflow for a catchment of that size. This affected less than 0.5 % of all data points. A similar procedure was used for low flows based on a dataset of the FOEN with the lowest recorded mean streamflows over seven days but this resulted in no replacements.

### 2.5.3 Temporal resolution of the observations

Data entries from citizen scientists are not as regular as data from sensors with a fixed temporal resolution. Therefore, we decided to test eight scenarios with a different temporal resolution and a different distribution of the data throughout the year to simulate different patterns in citizen contributions:

- *Hourly*: One data point per hour ($8760 \leq n \leq 8784$, depending on the year)
- *Weekly*: One data point per week, every Saturday, randomly between 6 am and 8 pm ($52 \leq n \leq 53$)
- *Monthly*: One data point per month on the 15th of the month, randomly between 6 am and 8 pm (n=12)
- *IntenseSummer*: One data point every other day from July until September, randomly between 6 am and 8 pm ($\sim$15 observations per month, n=46).
- *WeekendSummer*: one data point each Saturday and each Sunday between May and October, randomly between 6 am and 8 pm ($52 \leq n \leq 54$)
- *WeekendSpring*: one data point on each Saturday and each Sunday between March and August, randomly between 6 am and 8 pm ($52 \leq n \leq 54$)
- *Crowd52* had 52 random data points (in order to be comparable to the *Weekly*, *IntenseSummer*, *WeekendSummer* and *WeekendSpring* time series)
- *Crowd12* had 12 random data points (comparable to the *Monthly* data).

Except for the hourly data, these scenarios were based on our own experiences within the CrowdWater project (www.crowdwater.ch) and information from the CrowdHydrology project (Lowry and Fienen, 2013). The hourly dataset was included to test the effect of errors when the temporal resolution of the data is optimal (i.e., by comparing simulations for the models calibrated with the hourly FOEN data and those calibrated with hourly data with errors). In the two scenarios Crowd 52 and Crowd12 with random intervals between data points we assigned higher probabilities for periods when people are more likely to be outdoors (i.e., higher probabilities for summers than winters, higher probabilities for weekends than weekdays, higher probabilities outside office hours; Table 4). Times without daylight (dependent on the season) were always excluded. We used the same selection of days, including the same times of the day for each of the four different error groups, years and catchments to allow comparison of the different model results.

### 2.6 Model calibration

For each of the 1728 cases (6 catchments, 3 calibration years, 4 error groups, 8 temporal resolutions) the HBV model was calibrated by optimizing the overall consistency performance $P_{OA}$ (Finger et al., 2011) using a genetic optimization algorithm

(Seibert, 2000). The overall consistency performance $P_{OA}$ is the mean of four objective functions with an optimum value of one: *i*) NSE, *ii*) the NSE for the logarithm of streamflow, *iii*) the volume error, and *iv*) the mean absolute relative error (MARE). The parameters were calibrated within their typical ranges (see Supplemental Material - Table 1). To consider parameter uncertainty, the calibration was performed 100 times, which resulted in 100 parameter sets for each case. For each

case, the preceding year was used for the warm-up period. For the *Crowd52* and *Crowd12* time series, we used 100 different random selections of times for which data were available, whereas for the regularly spaced time series the same times were used for each case.

## 2.7   Model validation and analysis of the model results

**The 100 parameters from the calibration for each case were used to run the model for the validation years (**

Table 2). For each case (i.e. each catchment, year, error magnitude and temporal resolution), we determined the median validation $P_{OA}$ for the 100 parameter sets for each validation year. We analysed the validation results of all years combined and for all nine combinations of dry, mean and wet years separately.

Because the focus of this study was on the value of limited inaccurate streamflow observations for model calibration, i.e. the difference in the performance of the models calibrated with the synthetic data series compared to the performance of the

models calibrated with hourly FOEN data, all model validation performances are expressed relative to the average $P_{OA}$ of the model calibrated with the hourly FOEN data (our upper benchmark, representing the fully informed case when continuous high quality streamflow data are available). A relative $P_{OA}$ of 1 indicates that the model performance is as good as the performance of the model calibrated with the hourly FOEN data, whereas lower $P_{OA}$ values indicate a poorer performance.

In humid climates, the input data (precipitation and temperature) often dictate that model simulations can't be too far off as

long as the water balance is respected (Seibert et al., 2018). To assess the value of limited inaccurate streamflow data for model calibration compared to a situation without any streamflow data, a lower benchmark (Seibert et al., 2018) was used. Here, the lower benchmark was defined as the median performance of the model ran with 1000 random parameters sets. By running the model with 1000 randomly chosen parameter sets, we represent a situation where no streamflow data for calibration are available and the model is driven only by the temperature and precipitation data. We used 1000 different

parameter sets to cover most of the model variability due to the different parameter combinations. The Mann Whitney U-Test was used to evaluate whether the median $P_{OA}$ for a specific error group and temporal resolution of the data was significantly different from the median $P_{OA}$ for the lower benchmark (i.e. the model runs with random parameters). We furthermore checked for differences in model performance for models calibrated with the same data errors but different temporal resolutions using a Kruskal-Wallis test. By applying a Dunn-Bonferroni post-hoc test (Bonferroni, 1936; Dunn,

1959, 1961) we analysed which of the validation results were significantly different from each other.

The random generation of the 100 crowdsourced-like datasets (i.e. for the *Crowd52* and *Crowd12* scenario) for each of the catchments and year characteristics resulted in time series with a different number of high flow estimates. In order to find out whether the inclusion of more high flow values resulted in a better validation performance, we defined the threshold for high

flows as the streamflow value that was exceeded 10 % of the time in the hourly FOEN streamflow dataset. The *Crowd52* and *Crowd12* datasets were then divided into a group that had more than the expected 10 % high flow observations and a group that had fewer high flow observations. To determine if more high flow data improves model performance, the Mann-Whitney-U-test was used to compare the relative median $P_{OA}$ of the two groups.

## 3    Results

### 3.1    Upper benchmark results

The model was able to reproduce the measured streamflow reasonably well when the complete and unchanged hourly-FOEN datasets were used for calibration, although there were also a few exceptions. The average validation $P_{OA}$ was 0.61 (range: 0.19 – 0.83; Table 3). The validation performance was poorest for the Guerbe (validation $P_{OA} = 0.19$) because several high flow peaks were missed or underestimated by the model for the wet validation year. Similarly, the validation for the Mentue for the dry validation year resulted in a low $P_{OA}$ (0.23) because a very distinct peak at the end of the year was missed and summer low flows were overestimated. The third lowest $P_{OA}$ value was also for the Guerbe (dry validation year) but already had a $P_{OA}$ of 0.35. Six out of the nine lowest $P_{OA}$ values were for dry validation years. Validation for wet years for the models calibrated with data from wet years resulted in the best validation results (i.e., highest $P_{OA}$ values; Table 3).

### 3.2    Effect of errors on the model validation results

Not surprisingly, increasing the errors in the streamflow data used for model calibration led to a decrease in the model performance (Figure 4). For the small error category, the median validation performance was better than the lower benchmark for all temporal resolutions (Figure 4 and Supplemental Material - Table 2). For the medium error category, the median validation performance was also better than the lower benchmark for all scenarios, except for the *Crowd12* dataset. For the model calibrated with the dataset with large errors only the *Hourly* data set was significantly better than the lower benchmark (Table 5).

### 3.3    Effect of the data resolution on the model validation results

The *Hourly* measurement scenario resulted in the best validation performance for each error group, followed by the *Weekly* data, and then usually the *Crowd52* data (Figure 4). Although the median validation performance of the models calibrated with the *Weekly* datasets was better than for the *Crowd52* dataset for all error cases, the difference was only statistically significant for the no error category (Figure 5).

The validation performance of the models calibrated with the *Weekly* and *Crowd52* datasets was better than for the scenarios focused on spring and summer observations (*WeekendSpring*, *WeekendSummer* and *IntenseSummer*). The median model performance for the *Weekly* dataset was significantly better than the datasets focusing on spring and summer for the no, small and medium error groups. The median performance of the *Crowd52* dataset was only significantly better than all three measurement scenarios focusing on spring or summer for the small error case (Figure 5). The model validation performance

for the *WeekendSummer* and *IntenseSummer* scenarios decreased faster with increasing errors compared to the *Weekly*, *Crowd52* or *WeekendSpring* datasets (Figure 5). The median validation $P_{OA}$ for the models calibrated with the *WeekendSpring* observations was better than for the models calibrated with the *WeekendSummer* and *IntenseSummer* datasets but the differences were only significant for the small, medium and large error groups. The differences in the model

performance results for the observation strategies focussed on summer (*IntenseSummer* and *WeekendSummer*) were not significant for any of the error groups (Figure 5).

The median model performance for the regularly spaced *Monthly* datasets with 12 observations was similar to the median performance for the three datasets focusing on summer with 46-54 measurements (*WeekendSpring*, *WeekendSummer* and *IntenseSummer*), except for the case of large errors for which the monthly dataset performed worse. The irregularly spaced

*Crowd12* time series resulted in the worst model performance for each error group but the difference from the performance for the regularly spaced *Monthly* data was only significant for the dataset with large errors.

### 3.4    Effect of errors and data resolution on the parameter ranges

For most parameters the spread in the optimized parameter values was smallest for the upper benchmark. The spread in the parameter values increased with increasing errors in the data used for calibration, particularly for MAXBAS (the routing

parameter) but also for some other parameters (e.g. TCALT, TT and BETA). However, for some parameters (e.g., CFMAX, FC, and SFCF) the range in the optimized parameter values was mainly affected by the temporal resolution of the data and the number of data points used for calibration. It should be noted though that the changes in the range of model parameters differed significantly for the different catchments and the trends weren't very clear.

### 3.5    Influence of the calibration and validation year and number of high flow data points on the model

20       **performance**

The influence of the validation year on the model performance was larger than the effect of the calibration year (Figure 6 and Supplemental Material – Figure 2). In general model performance was poorest for the dry validation years. The model performances of all datasets with fewer observations or bigger errors than the *Hourly* datasets without errors were not significantly better than the lower benchmark for the dry validation years, except for *Crowd52* in the no error group when

calibrated with data from a wet year. However, even for the wet validation years some observation scenarios of the no error and small error group did not lead to significantly better model validation results compared to the median validation performance for the random parameters. Interestingly, the *IntenseSummer* data set in the no error group resulted in a very good performance when the model was calibrated for a dry and also validated in a dry year compared to its performance in the other calibration and validation year combinations. The median model performance was however not significantly better

than the lower benchmark due to the low performances for the Guerbe and Allenbach (outliers beyond figure margins in Figure 6). The validation results of one of these two catchments always resulted in the worst performance for all the no error - *IntenseSummer* datasets for all calibration and validation year combinations.

For 13 out of the 18 catchment and year combinations, the Crowd52 datasets with fewer than 10% high streamflow data points led to a better validation performance than the Crowd52 datasets with more high streamflow data points. For six of them the difference in model performance was significant. For none of the five cases where more high flow data points led to a better model performance was the difference significant. Also when the results were analysed by year character or catchment there was no improvement when more high flow values were included in the calibration dataset.

## 4 Discussion

### 4.1 Usefulness of inaccurate streamflow data for hydrological model calibration

In this study, we evaluated the information content of streamflow estimates by citizen scientists for calibration of the bucket-type hydrological model for six Swiss catchments. Streamflow estimates by citizens are sometimes very differ-ent from the measured values, and the individual estimates can be dis-informative for model calibration (Beven, 2016; Beven and Westerberg, 2011). While the hydroclimatic conditions, the model or the calibration approaches might be different in other studies, these results should be applicable for a wide range of cases. However, for physically-based spatially distributed models that are usually not calibrated automatically, the use of limited streamflow data would probably benefit from a different calibration approach. Furthermore, our results might not be applicable in arid catchment cases where rivers fall dry for some period of the year because the linear reservoirs used in the HBV model are not appropriate for such systems. The results show that if the streamflow estimates by citizen scientists would be available at a high temporal resolution (hourly), these data are informative for the calibration of a bucket-type hydrological model despite their high uncertainties. However, observations with such a high resolution are very unlikely to be obtained in practice. All scenarios with error distributions that represent the estimates from citizen scientists with fewer observations were no better than the lower benchmark (using random parameters). With medium errors, however, and one data point per week on average or regularly spaced monthly data, the data were informative for model parameterization. Reducing the standard deviation of the error-distribution by a factor of four, led to significantly improved model performance for all the observation scenarios compared to the lower benchmark.

A reduction in the errors of the streamflow estimates could be achieved by training of citizen scientists (e.g. videos), improved information about feasible ranges for stream depth, with and velocity, or examples of streamflow values for well-known streams. Filtering of extreme outliers can also reduce the spread of the estimates. This could be done with existing knowledge of feasible streamflow values for a catchment of a given area or the amount of rainfall right before the estimate is made to determine if streamflow is likely to be higher or lower than for the previous estimate. More detailed research is necessary to test the effectiveness of such methods.

Le Coz et al. (2014) reported an uncertainty in stage-discharge streamflow measurements of around 5-20 %. McMillan et al. (2012) summarized streamflow uncertainties from stage-discharge relationships in a more detailed review and gave a range of ±50-100 % for low flows, ±10-20 % for medium or high (in-bank) flows and ±40 % for out-of-bank flows. The errors for

the most extreme outliers in the citizen estimates are considerably higher, as they can differ by a factor of up to 10'000 from the measured value in the most extreme but rare cases (Figure 2). Even with reduced standard deviations of the error distribution by a factor of two or four, the observations in the most extreme cases can still differ by a factor of 100 and 10. The percentage of values beyond 200 % of the measured value in the synthetic datasets with streamflow observations was 33 % for the large error group, 19 % in the medium error group and 4 % in the small error group. Only 3 % were more than 90 % below the measured value in the large error group and 0 % for both in the medium and small error classes. If such observations are used for model calibration without filtering, they are seen as extreme droughts or floods, even if the actual conditions may be close to average flow. Beven and Westerberg (2011) suggest to isolate periods of dis-informative data. It is therefore beneficial to identify such extreme outliers, independent of a model, e.g. with knowledge of feasible maximum and minimum streamflow quantities, as used in this study, with the help of the maximum regionalized specific streamflow values for a given catchment area.

## 4.2    Number of streamflow estimates required for model calibration

In general, one would assume that the calibration of a model becomes better when there is more data (Perrin et al., 2007), although others have shown that the increase in model performance plateaus after a certain number of measurements (Juston et al., 2009; Pool et al., 2017; Seibert and Beven, 2009; Seibert and McDonnell, 2015). In this study, we limited the length of the calibration period to one year because in practice it may be possible to obtain a limited number of measurements during a one year period for ungauged catchments before the model results are needed for a certain application, as has been assumed in previous studies (Pool et al., 2017; Seibert and McDonnell, 2015). While a limited number of observations (12) was informative for model calibration when the data uncertainties were limited, the results of this study also suggest that the performance of bucket-type models decreases faster with increasing errors when fewer data points are available (i.e. there was a faster decline in model performance with increasing errors for models calibrated with 12 data points than for the models calibrated with 48-52 data points). This finding was most pronounced when comparing the model performance for the small and the medium error groups (Figure 4). These findings can be explained by the compensating effect of the number of observations and their accuracy because the random errors for the inaccurate data average out when a large number of observations are used, as long as the data do not have a large bias.

## 4.3    Best timing of streamflow estimates for model calibration

The performance of the parameter sets depended on the timing and the error distribution of the data used for model calibration. The model performance was generally better if the observations were more evenly spread throughout the year. For example for the cases of no and small errors, the performance of the model calibrated with the *Monthly* dataset with 12 observations was better than for the *IntenseSummer* and *WeekendSummer* scenarios with 46-54 observations. Similarly, the less clustered observation scenarios performed better than the more clustered scenarios (i.e. *Weekly* vs. *Crowd52*, *Monthly* vs. *Crowd12*, *Crowd52* vs. *IntenseSummer,* etc.). This suggests that more regularly distributed data over the year leads to a

better model calibration. Juston et al. (2009) compared different subsamples of hydrological data for a 5.6 km$^2$ Swedish catchment and found that including inter-annual variability in the data used for the calibration of the HBV model reduced the model uncertainties. More evenly distributed observations throughout the year might represent more of the within-year streamflow variability and therefore result in improved model performance. This is good news for using citizen science data for model calibration as it suggests that the timing is not as important as the number of observations because it is likely much easier to get observations throughout the year than during specific periods or flow conditions.

When comparing the *WeekendSpring*, *WeekendSummer* and *IntenseSummer* datasets, it seems that it was in most cases more beneficial to include data from spring rather than summer. This tendency was more pronounced with increasing data errors. The reason for this might be that the *WeekendSpring* scenario includes more snow melt or rain-on-snow event peaks, in addition to usually higher baseflow and therefore contains more information on the inter-annual variability in streamflow.

By comparing different variations of 12 data points to calibrate the HBV model, Pool et al. (2017) found that a dataset that contains a combination of different maximum (monthly, yearly etc.) and other flows in model calibration led to the best model performance but also that the differences between the different datasets covering the range of flows were small. In our study we did not specifically focus on the high or low flow data points, and therefore did not have datasets that contained only high flow estimates, which would be very difficult to obtain with citizen science data. However, our findings similarly show that for model calibration for catchments with seasonal variability in streamflow it is beneficial to obtain data for different magnitudes of flow. Furthermore, we found that data points during relatively dry periods are beneficial for validation or prediction in another year and might even be beneficial for years with the same characteristics, as was shown for the improved validation performance of the *IntenseSummer* dataset compared to the other datasets when data from dry years were used for calibration (Figure 6).

## 4.4    Effects of different types of years on model calibration and validation

The calibration year, i.e. the year in which the observations were made, was not decisive for the model performance. Therefore, a model calibrated with data from a dry year can still be useful for simulations for an average or wet years. This also means that data in citizen science projects can be collected during any year and that this data is useful for simulating streamflow for most years, except the driest years. However, model performance did vary significantly for the different validation years. The results during dry validation years were almost never significantly better than the lower benchmark (Supplemental Material – Figure 2). This might be due to the objective function that was used in this study. Especially the NSE was lower for dry years, because the flow variance (i.e., the denominator in the equation) is smaller when there is a larger variation in streamflow. Also, these results are based on six median model performances and therefore, outliers have a big influence on the significance of results (Supplemental Material – Figure 2).

Lidén and Harlin (2000) used the HBV-96 model by Lindström et al. (1997) with changes suggested by Bergström et al. (1997) for four catchments in Europe, Africa and South America. They achieved better model results for wetter catchments and argued that during dry years evapotranspiration plays a bigger role and therefore the model performance is more

sensitive to inaccuracies in the simulation of the evapotranspiration processes. The fact that we used a very simple method to calculate the potential evapotranspiration (McGuinness and Bordne, 1972), might also explain why the model performed less well during dry years.

The model parametrisation, obtained from calibration using the *IntenseSummer* data set resulted in a surprisingly good performance for the validation for a more extreme dry year for four out of the six catchments. For the two catchments for which the performance for the *IntenseSummer* data set was poor (Guerbe and Allenbach), the weather stations are located outside the catchment boundaries. Especially during dry periods missed streamflow peaks due to misrepresentation of precipitation can affect model performance a lot. The fact that always one of these two catchments had the worst model performance for all the no error – *IntenseSummer* runs, furthermore indicates that the July-September period might not be suitable to represent characteristic runoff events for these catchments. The bad performance for these two catchments for the *IntenseSummer* – no error run with calibration and validation in the dry year resulted in the insignificant improvement in model performance compared to the lower benchmark. Because the wetness of a year was based on the summer streamflow, these findings suggest that data obtained during times of low flow, result in improved validation performance during dry years compared to data collected during other times (Supplemental Material – Figure 2). This suggests that if the interest is in understanding the streamflow response during very dry years, it is important to obtain data during the dry period. To test this hypothesis more detailed analyses are needed.

### 4.5 Recommendations for citizen science projects

Our results show that streamflow estimates from citizens are not informative for hydrological model calibration, unless the errors in the estimates can be reduced through training or advanced filtering of the data to reduce the errors (i.e. to reduce the number of extreme outliers). In order to make streamflow estimates useful, the standard deviation of the estimation-error-distribution needs to be reduced by a factor of two. Gibson and Bergman (1954) suggest that errors in distance estimates can be reduced from 33 % to 14 % with very little training. These findings are encouraging, although their tests covered distances larger than 365 meters (400 yards) and the widths of the medium sized rivers for which the streamflow was estimated were less than 40 meters (Strobl et al., in review). Options for training might be tutorial videos, as well lists with values for the width, average depth and flow velocity of well-known streams (Strobl et al., in review). In order to determine the effect of training on streamflow estimates further research has to be done because especially the depth estimates were inaccurate (Strobl et al., in review).

The findings of this study suggest the following recommendations for citizen science projects that want to use streamflow estimates:

- Collect as much data as possible: In this study hourly data always led to the best model performance. It is therefore beneficial to collect as much data as possible. Because it is unlikely to obtain hourly data, we suggest to aim for (on average) one observation per week. Provided that the standard deviation of the streamflow estimates can be reduced by a factor of two, 52 observations (as in the *Crowd52* data series) are informative for model calibration. Therefore,

it is essential to invest in advertisement of a project and to find suitable locations where many people can potentially contribute, as well as to communicate to the citizen scientists that it is beneficial to submit observations regularly.

- Encourage observations throughout the year: To further improve the model performance, or to allow for greater errors, it is beneficial to have observations at all types of flow conditions during the year, rather than during a certain season.

Observations during high streamflow conditions were in most cases not more informative than flows during other times of the year. Efforts to ask citizens to submit observations during specific flow conditions (e.g. by sending reminders to the citizen observers) do not seem very effective in light of the above findings. It is rather more beneficial to remind them to submit observations regularly.

Instead of focussing on training to reduce the errors in the streamflow estimates, an alternative approach for citizen science projects is to switch to a parameter that is easier to estimate, such as stream levels (Lowry and Fienen, 2013). Recent studies successfully used daily stream level data (Seibert and Vis, 2016) and stream level class data (van Meerveld et al. 2017) to calibrate hydrological models, and other studies demonstrated the potential value of crowdsourced stream level data for providing information on e.g. baseflow (Lowry and Fienen, 2013) or to improve flood forecasts (Mazzoleni et al., 2017). However, further research is needed to determine if real crowdsourced stream level (class-) data is informative for the calibration of hydrological models.

## 5    Conclusions

The results of this study extend previous studies on the value of limited hydrological data for hydrological model calibration or the best timing of streamflow measurements for model calibration (Juston et al., 2009; Pool et al., 2017; Seibert and McDonnell, 2015) that did not consider observation errors. This is an important aspect, especially when considering citizen science approaches to obtain streamflow data. Our results show that inaccurate streamflow data can be useful for model calibration, as long as the errors are not too large. When the distribution of errors in the streamflow data represented the distribution of the errors in the streamflow estimates from citizen scientists, this information was not informative for model calibration (i.e. the median performance of the models calibrated with these data was not significantly better than the median performance of the models with random parameter values). However, if the standard deviation of the estimates is reduced by a factor two, then the (less) inaccurate data would be informative for model calibration. We, furthermore, demonstrated that realistic frequencies for citizen science projects (one observation on average per week or month) can be informative for model calibration. The findings of studies such as the one presented here provide important guidance on the design of citizen science projects, and also other, observation approaches.

## 6 Author contribution

While Jan Seibert and Ilja van Meerveld had the initial idea, the concrete study design was based on input from all authors. Simon Etter and Barbara Strobl conducted the field surveys to determine the typical errors in the streamflow estimates. The simulations and analyses were performed by Simon Etter. The writing of the manuscript was led by Simon Etter; all co-authors contributed to the writing.

## 7 Data availability

The data are available from FOEN (streamflow) and MeteoSwiss (precipitation and temperature). The HBV software is available from https://www.geo.uzh.ch/en/units/h2k/Services/HBV-Model.html.

## 8 Acknowledgements

We thank all citizen scientists who participated in the field surveys, as well as the Swiss Federal Office for the Environment for providing the streamflow data, MeteoSwiss for providing the weather data, Maria Staudinger, Jan Schwanbeck and Scherrer AG for the permission to use their datasets, and the reviewers for the useful comments. This project was funded by the Swiss National Science Foundation (project CrowdWater).

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

**Tables**

**Table 1 Characteristics of the six Swiss catchments used in this study. For the location of the study catchments see Figure 1. Long-term averages are for the period 1974-2014, except for Verzasca for which the long term average is for the 1990-2014 period. Regime types are classified according to (Aschwanden and Weingartner, 1985).**

| Catchment | | Murg | Guerbe | Allenbach | Riale di Calneggia | Mentue | Verzasca |
|---|---|---|---|---|---|---|---|
| **Gauging station (FOEN station number)** | | Waengi (2126) | Belp Mülimatt (2159) | Adelboden (2232) | Cavergno, Pontit (2356) | Yvonand La Mauguettaz (2369) | Lavertezzo, Campiòi (2605) |
| **Area [km$^2$]** | | 79 | 117 | 29 | 24 | 105 | 186 |
| **Elevation [m asl]** | Min | 465 | 522 | 1297 | 885 | 445 | 490 |
| | Max | 1035 | 2176 | 2762 | 2921 | 927 | 2864 |
| **Regime Type** | | Pluvial-inférieur | Pluvial-superieur | Nival-alpin | Nival-méridional | Pluvial-jurassien | Nivo-pluvial-méridional |
| **Min - Max Pardé coefficients** | Dry year | 0.29 - 1.61 | 0.44 - 1.93 | 0.40 - 2.48 | 0.13 - 3.22 | 0.22 - 2.37 | 0.16 - 2.92 |
| | Average year | 0.58 - 2.16 | 0.61 - 1.65 | 0.39 - 2.44 | 0.09 - 2.84 | 0.23 - 2.66 | 0.23 - 3.17 |
| | Wet year | 0.34 - 1.69 | 0.42 - 2.14 | 0.32 - 2.12 | 0.10 - 3.48 | 0.35 - 2.39 | 0.26 -2.64 |
| | Long-term | 0.68 - 1.34 | 0.77 - 1.39 | 0.35 - 2.70 | 0.14 - 2.70 | 0.46 - 1.57 | 0.23 - 2.22 |
| **Annual runoff-rainfall ratio** | Dry year | 0.72 | 0.37 | 0.86 | 1.30[1] | 0.41 | 0.98 |
| | Average year | 0.55 | 0.48 | 1.73[1] | 1.38[1] | 0.52 | 0.66 |
| | Wet year | 0.56 | 0.54 | 0.78 | 0.98 | 0.50 | 1.32[1] |
| | Long-term | 0.56 | 0.57 | 0.94 | 1.06[1] | 0.38 | 0.9 |
| **Long-term mean annual streamflow [m$^3$/s]** | | 1.84 | 2.75 | 1.23 | 1.43 | 1.64 | 10.76 |
| **Weather stations** | | Aadorf-Taenikon, Hörnli | Plaffeien, Bern-Zollikofen | Adelboden | Robiei | Mathod, Pully | Acquarossa, Cimetta, Magadino, Piotta |

---

[1] In Verzasca, Allenbach,and Riale die Calneggia there are some streamflow-rainfall ratios >1 because the weather stations are located outside the catchment and precipitation is highly variable in this alpine terrain.

**Table 2** Calibration years (2nd-most extreme and 2nd-closest to average years) and validation years (most extreme and closest to average years) for each catchment. Numbers in parenthesis are the ranks over the period 1974-2014 (or 1990-2014 for Verzasca).

| Year character | Murg | Guerbe | Allenbach | Riale di Calneggia | Mentue | Verzasca |
|---|---|---|---|---|---|---|
| | | | **Calibration** | | | |
| **Wet** | 2007 (3) | 2007 (2) | 2007 (4) | 2009 (11) | 2014 (7) | 2011 (4) |
| **Dry** | 2013 (8) | 2011 (8) | 2009 (11) | 2012 (8) | 2010 (4) | 2013 (5) |
| **Average** | 2008 (6) | 2008 (17) | 2013 (7) | 2013 (2) | 2006 (6) | 2007 [7] |
| | | | **Validation** | | | |
| **Wet** | 2014 [1] | 2014 [1] | 2014 [1] | 2008 [9] | 2007 [1] | 2008 [1] |
| **Dry** | 2009 (7) | 2013 (5) | 2012 (9) | 2006 (5) | 2009 (3) | 2010 (4) |
| **Average** | 2011 (4) | 2006 (13) | 2011 (6) | 2011 (1) | 2013 (2) | 2006 (4) |

**Table 3** Median and the full range of $P_{OA}$ scores for the upper benchmark (hourly-FOEN data). The upper benchmark values for the dry, average and wet calibration years were used as the upper benchmarks for the evaluation based on the year character (Figure 6 and Supplemental Material – Figure 2); the values in the "overall median"-column were used as the benchmarks in the overall median performance evaluation shown in Figure 4.

| Calibration year | Dry | Average | Wet | Overall median |
|---|---|---|---|---|
| | **Validation wet year** | | | |
| **Upper benchmark** | 0.63 (0.19 - 0.79) | 0.65 (0.36 - 0.8) | 0.66 (0.45 - 0.8) | |
| **Lower benchmark** | 0.34 (-0.02 - 0.47) | | | |
| | **Validation average year** | | | **Upper benchmark** 0.61 (0.19 - 0.83) |
| **Upper benchmark** | 0.59 (0.49 - 0.64) | 0.61 (0.45 - 0.78) | 0.53 (0.36 - 0.77) | |
| **Lower benchmark** | 0.36 (0.03 - 0.59) | | | **Lower benchmark** 0.34 (-0.02 - 0.59) |
| | **Validation dry year** | | | |
| **Upper benchmark** | 0.51 (0.35 - 0.71) | 0.59 (0.41 - 0.83) | 0.53 (0.23 - 0.74) | |
| **Lower benchmark** | 0.35 (0.09 - 0.52) | | | |

**Table 4** Weights assigned to specific seasons, days and times of the day for the random selection of data points for *Crowd52* and *Crowd12*. The weights for each hour were multiplied and normalized. We then used them as probabilities for the individual hours. For times without daylight the probability was set to zero.

| Variable | | Weight |
|---|---|---|
| *Season* | | |
| December – February | | 2 |
| March – May / September – November) | | 6 |
| June – August) | | 10 |
| *Day* | | |
| Saturdays – Sundays | | 3 |
| Monday – Friday | | 1 |
| *Time* | | |
| Times when people usually have breaks | 6 am – 8:00 am, 12 am-1 pm, 5 pm-9 pm | 3 |
| Times with daylight in winter (Dec-Feb) | 8 am – 4 pm | 1 |
| Times with daylight in spring/fall (Mar-May/Sept-Nov): | 7 am – 7 pm | 1 |
| Times with daylight in summer (Jun-Aug) | 6 am – 9 pm | 1 |
| Other times (depending on season) | | 0 |

**Figures**

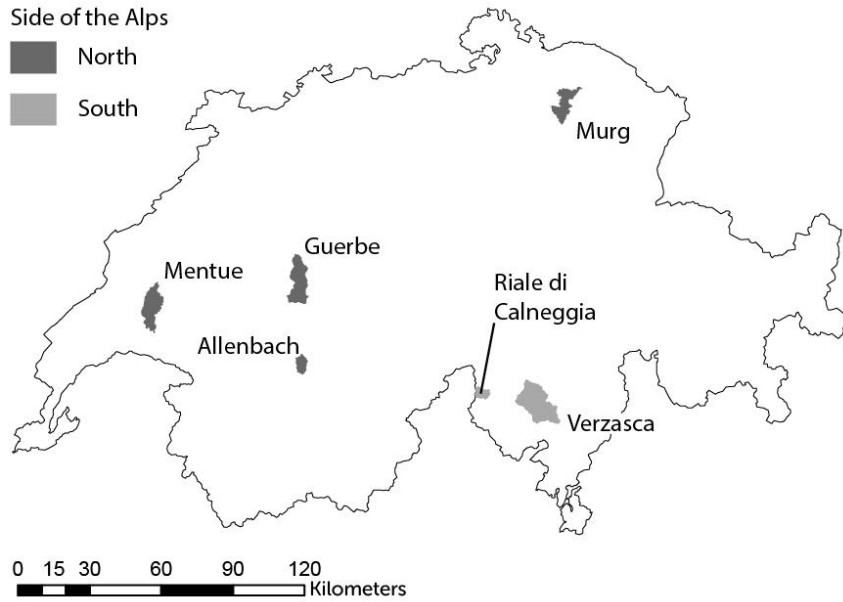

**Figure 1 Location of the six study catchments in Switzerland. Shading indicates whether the catchment is located on the north or south side of the Alps. See Table 1 for the characteristics of the study catchments.**

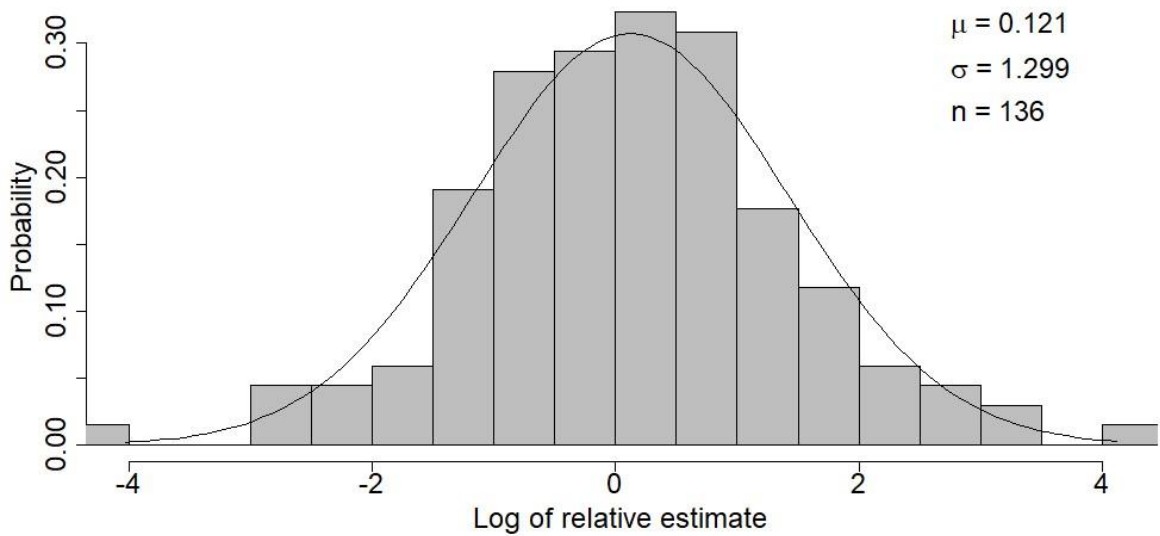

**Figure 2 Fit of the normal distribution to the frequency distribution of the log transformed relative streamflow estimates (ratio of the estimated streamflow and the measured streamflow).**

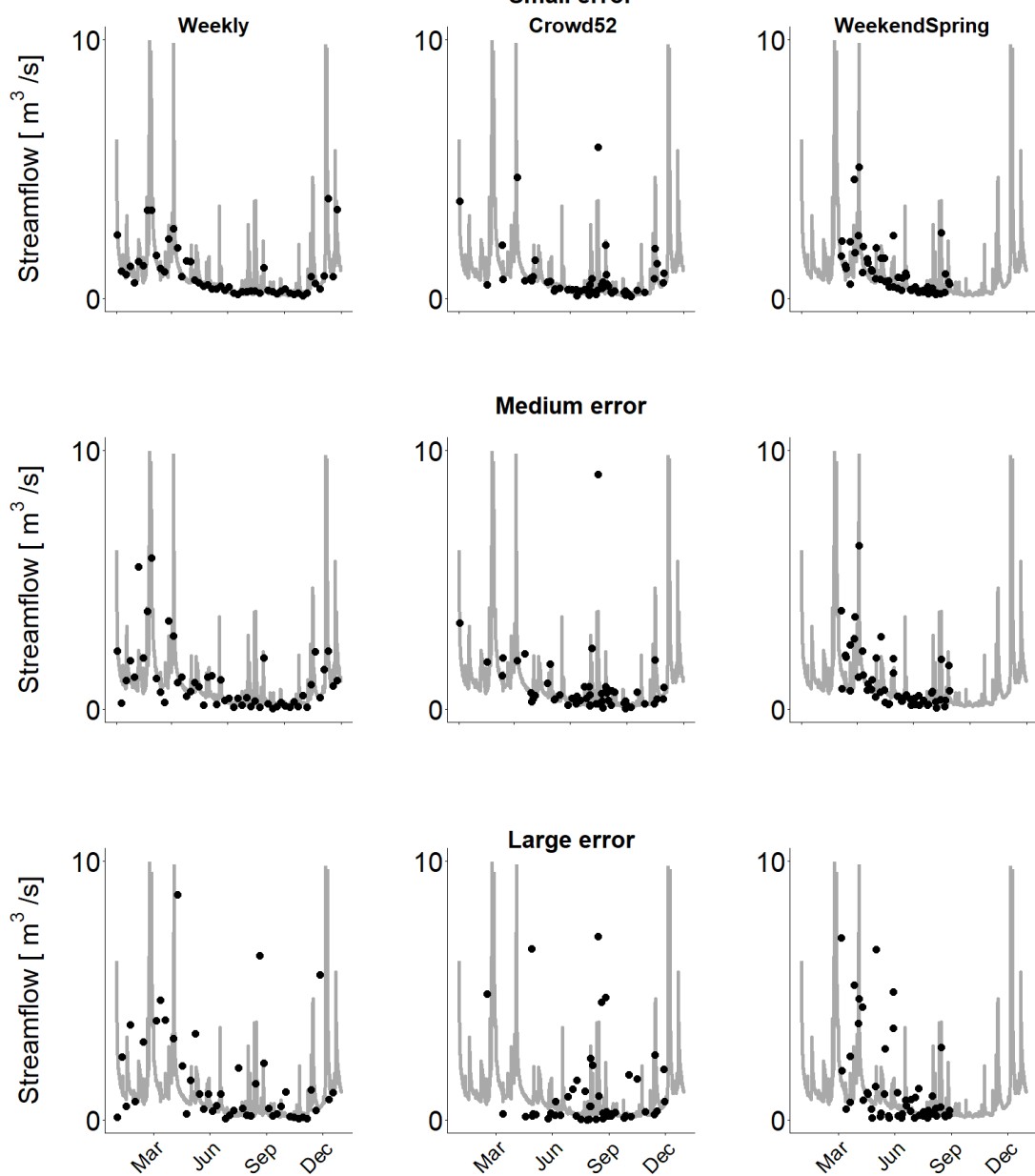

**Figure 3 Example of different streamflow time series used for calibration with small, medium and large errors and the temporal resolutions (*Weekly*, *Crowd52* and *WeekendSpring*) for the Mentue in 2010. Large error: adjusted FOEN data with errors resulting from the log-normal distribution fitted to the streamflow estimates from citizen scientists (see Figure 2). Medium error: same as large error, but the standard deviation of the log normal distribution was divided by 2. Small error: same as the large error, but the standard deviation of the log normal distribution was divided by 4. The grey line represents the measured streamflow, the dots**

the derived time series of streamflow observations. Note that especially in the large error category some dots lie outside the figure margins.

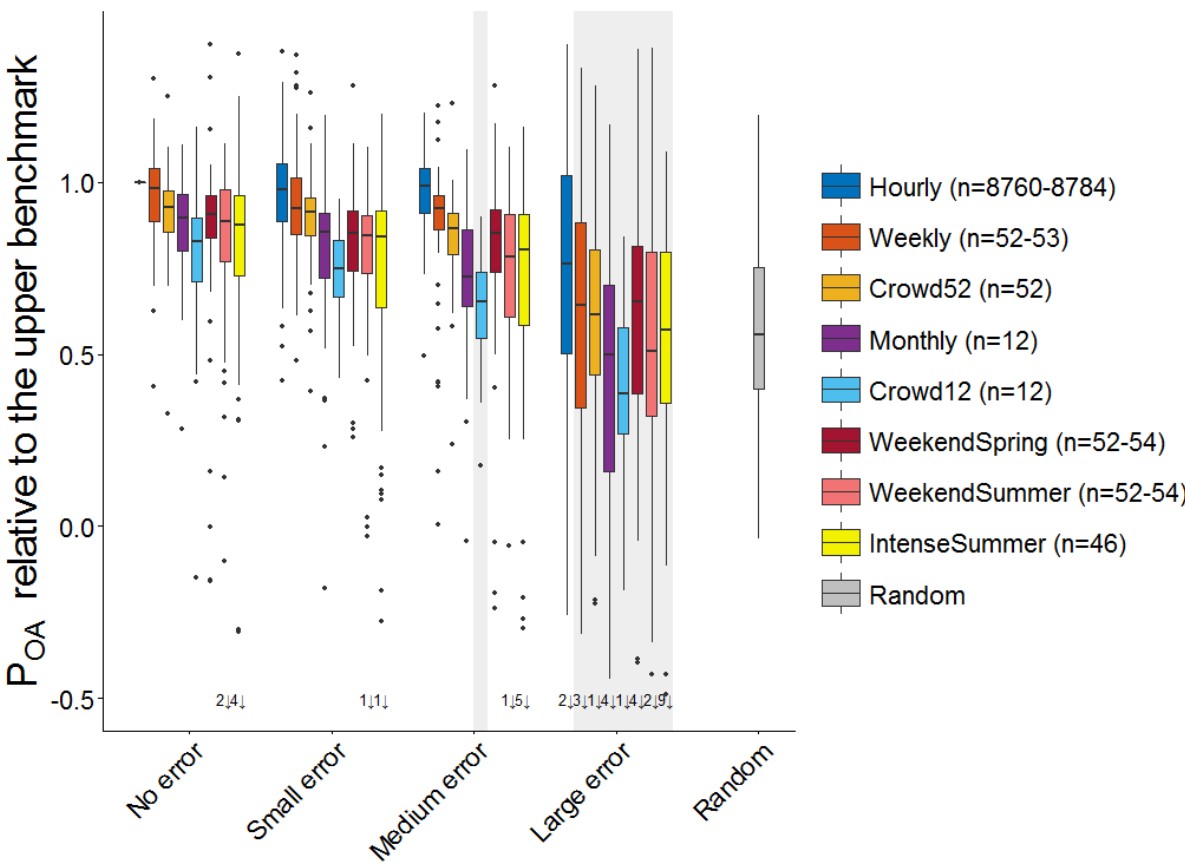

Figure 4 Boxplots of the median model performance relative to the upper benchmark for all datasets. The grey rectangles around the boxes indicate non-significant differences in median model performance compared to the lower benchmark with random parameter sets. The box represents the 25th and 75th percentile, the thick horizontal line the median, the whiskers extend to 1.5 times the interquartile range below the 25thpercentile and above the 75th percentile, and the dots represent the outliers. The

10  numbers at the bottom indicates the number of outliers beyond the figure margins. *n* is the number of streamflow observations used for model calibration. The result of the hourly-benchmark FOEN dataset has some spread because the results of the 100 parameters sets were divided by their median performance. A relative $P_{OA}$ of 1 indicates that the model performance is as good as the performance of the model calibrated with the hourly FOEN data (upper benchmark).

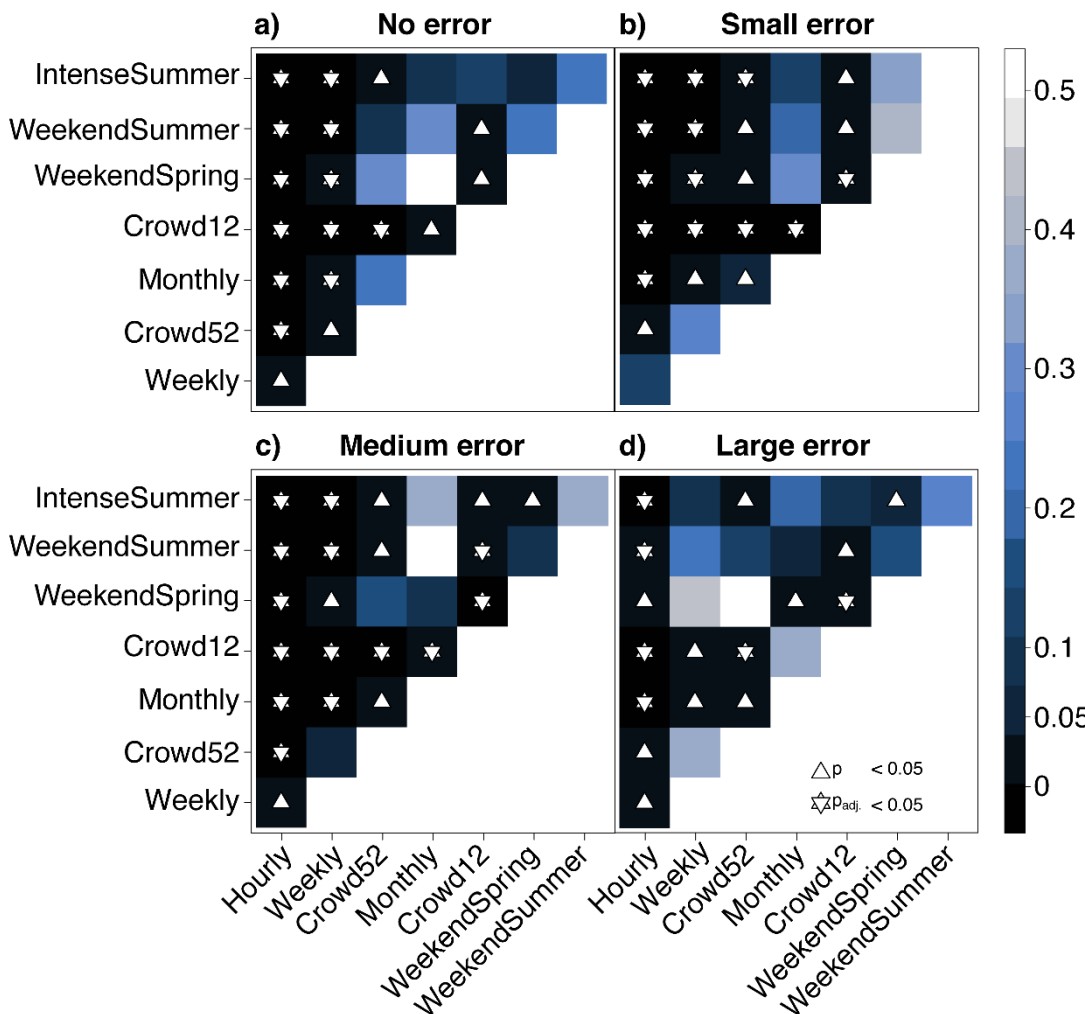

**Figure 5 Results (p-values) of the Bonferroni Post-Hoc test to determine the significance of the difference in the median model performance for the data with different temporal resolutions within each data quality group (no error (a), small error (b), medium error (c), and large error (d)). Blue shades represent the p-values. White triangles indicate p-values < 0.05 and white stars indicate p-values that, when adjusted for multiple comparisons, are still < 0.05.**

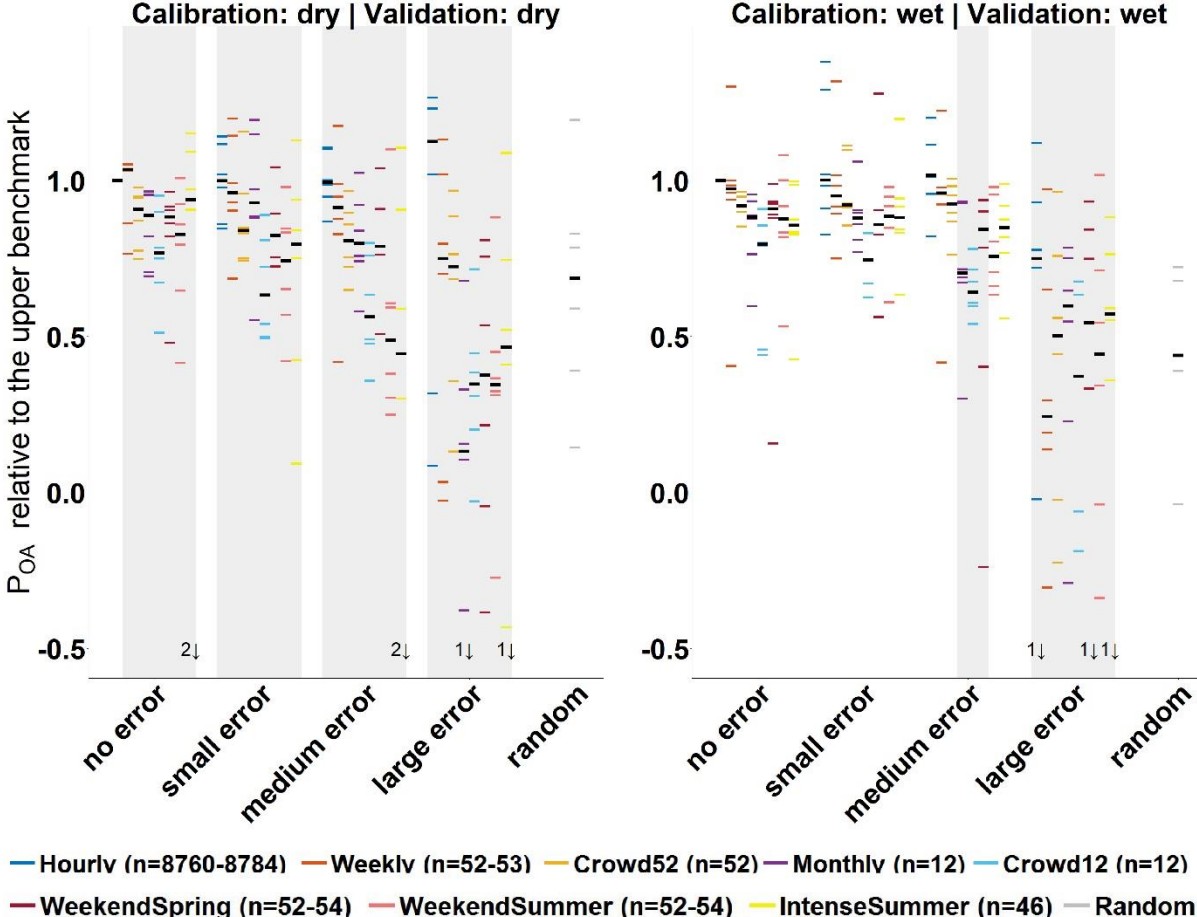

Figure 6 Median model validation performance for the datasets calibrated and validated both in a dry year and both in a wet year. Each horizontal line represents the median model performance for one catchment. The black bold line represents the median for the six catchments. The grey rectangles around the boxes indicate non-significant differences in median model performance for the six catchments compared to the lower benchmark with random parameters. The numbers at the bottom indicate the number of outliers beyond the figure margins. For the individual $P_{OA}$ values of the upper benchmark (no error – *Hourly* dataset) in the different calibration and validation years see Table 3.

11 **Supplemental Material**

12 **Model parameters**

13 **Supplemental Material - Table 1 Parameter ranges used for calibration of the HBV-model**

| Parameter | Description[a] | Unit | Min | Max |
|---|---|---|---|---|
| Rescaling Parameters of Input Data | | | | |
| PCALT | change in precipitation with elevation | % (100m)$^{-1}$ | 5 | 15 |
| TCALT | change in temperature with elevation | °C (10m)$^{-1}$ | 0.5 | 1.5 |
| Snow and ice melt parameters | | | | |
| TT | threshold temperature for liquid and solid precipitation | °C | -3 | 1 |
| CFMAX | degree-day factor | mmd$^{-1}$°C$^{-1}$ | 0.06 | 10 |
| SFCF | snowfall correction factor | - | 0.4 | 1.6 |
| CFR | refreezing coefficient | - | 0.001 | 0.9 |
| CWH | water holding capacity of the snow storage | - | 0.001 | 0.9 |
| Soil Parameters | | | | |
| PERC | maximum percolation from upper to lower groundwater storage | mm d$^{-1}$ | 0 | 3 |
| UZL | threshold parameter | mm | 0 | 100 |
| K0 | storage (or recession) coefficient 0 | d$^{-1}$ | 0.001 | 0.5 |
| K1 | storage (or recession) coefficient 1 | d$^{-1}$ | 0.0001 | 0.2 |
| K2 | storage (or recession) coefficient 2 | d$^{-1}$ | 2E-06 | 0.005 |
| MAXBAS | length of triangular weighting function | H | 1 | 7 |
| FC | maximum soil moisture storage | Mm | 50 | 550 |
| LP | soil moisture value above which actual evapotranspiration reaches potential evapotranspiration | - | 0.3 | 1 |
| Beta | shape factor for the function used to calculate the distribution of rain and snow melt going to runoff and soil box, respectively | - | 1 | 5 |

[a] a detailed description of the model parameters is given in (Seibert and Vis, 2012).

15 **Significance of median model performance compared to the lower benchmark**

16 **Supplemental Material - Table 2 Significance of the differences in median model performance for each temporal**
17 **resolution and an error group compared to the lower benchmark (Mann-Whitney U-test). The p-values of the**
18 **Kruskal-Wallis test for the within group variability in the lowermost row shows that the median model performance**
19 **of the different error groups was significantly different.**

|  | No Error | Small Error | Medium Error | Large Error |
|---|---|---|---|---|
| Hourly | <0.01 | <0.01 | <0.01 | <0.01 |
| Weekly | <0.01 | <0.01 | <0.01 | 0.75 |
| Crowd52 | <0.01 | <0.01 | <0.01 | 0.40 |
| Monthly | <0.01 | <0.01 | <0.01 | 0.03* |
| Crowd12 | <0.01 | <0.01 | 0.11 | <0.01* |
| WeekendSpring | <0.01 | <0.01 | <0.01 | 0.40 |
| WeekendSummer | <0.01 | <0.01 | <0.01 | 0.46 |
| IntenseSummer | <0.01 | 0.01 | 0.04 | 0.21 |
| Within error group | <0.01 | <0.01 | <0.01 | <0.01 |

* These datasets result in significantly worse results than random parameters.

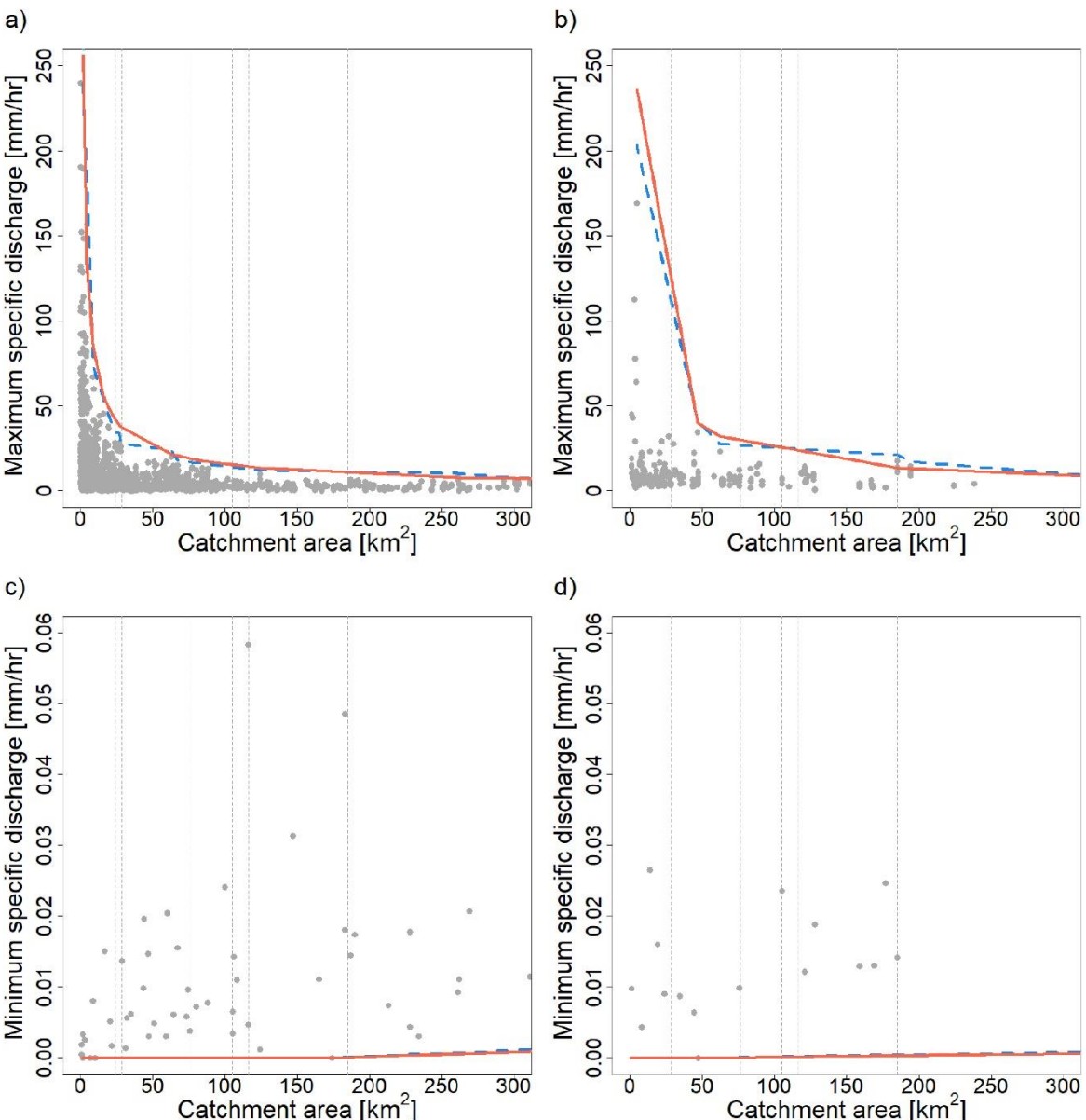

**Supplemental Material – Figure 1** Relation between catchment area and maximum (a, b) and minimum (c, d) specific streamflow for catchments on the north (a, c) and south (b, d) of the Alps. The dashed light blue line is the Pareto front including the 20 % buffer. The red lines are the fitted logarithmic models used to find the maximum and minimum possible flow for each catchment.

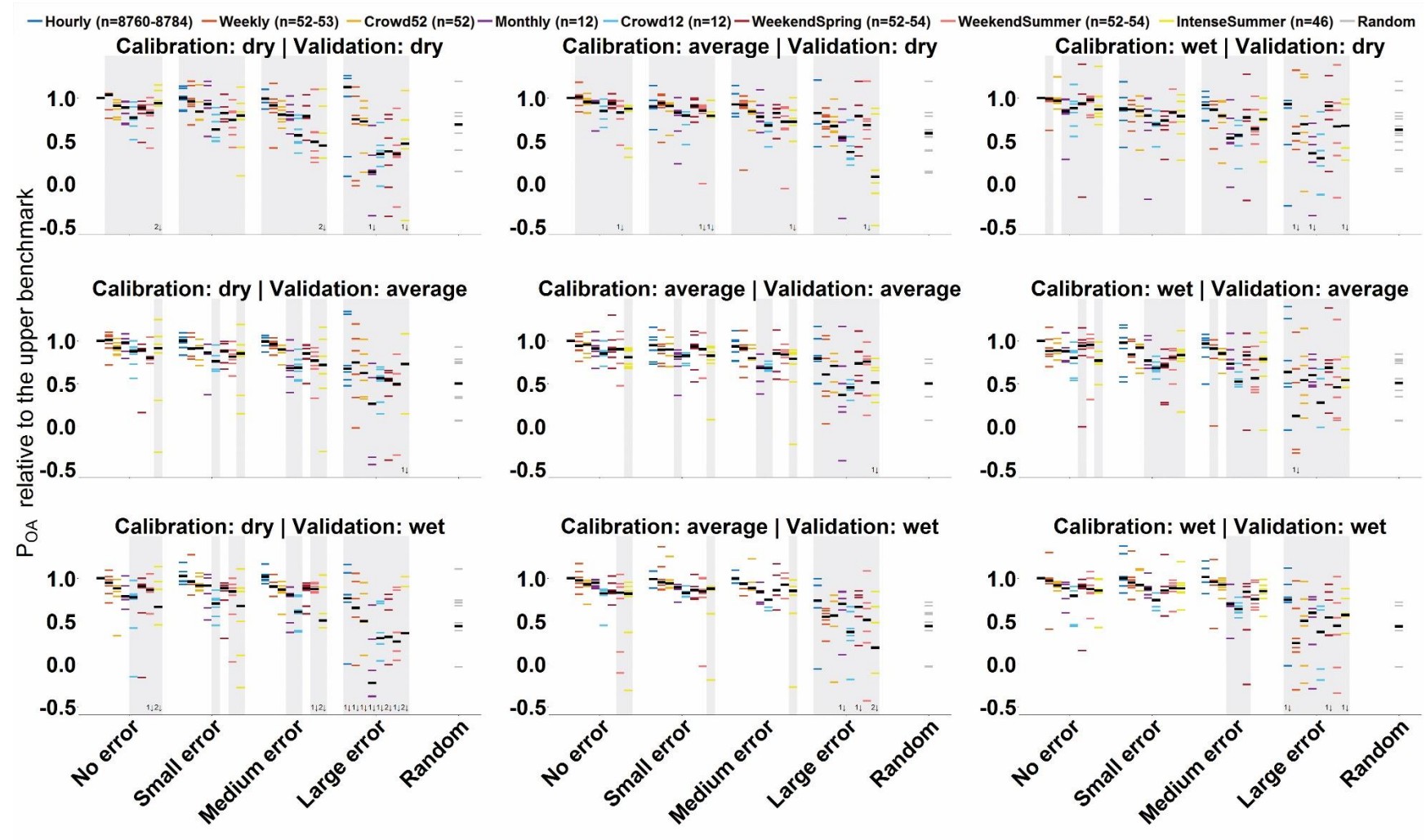

**Supplemental Material – Figure 2** Median model validation performance for all datasets used for calibration during the different validation periods. Each horizontal line represents the median model performance for one catchment. The black bold line represents the median for the six catchments. The grey rectangles around the boxes indicate non-significant differences in median model performance for the six catchments compared to the lower benchmark with random parameters. The numbers at the bottom indicate the number of outliers beyond the figure margins. For the individual $P_{OA}$ values of the upper benchmark (no error – *Hourly* dataset) in the different calibration and validation years see Table 3.