# Peer review of "Value of uncertain streamflow observations for hydrological modelling"

_Hydrology and Earth System Sciences, 2018_

## Referee Comment (RC1) · Anonymous Referee #1 · 13 Jul 2018

The manuscript entitled "Value of uncertain streamflow observations for hydrological modelling" presents interesting and novel research on the worth of citizen science discharge observations for the calibration of lumped hydrological models. The manuscript is well structured and concise with a clear motivation. The presentation and the application of the methods are scientifically sound. My comments are mostly of minor character and therefore I hope to see this article soon published in HESS.

general comments:

-The presentation of the calibration experiments is clear and complete with regard to the model performance. However, I was wondering how the model robustness is affected by uncertain observations which was neglected by the authors. From a modelling point of view parameter uncertainty and its reduction through calibration is of

high importance. Therefore I believe that an additional figure on that matter would improve the quality of the study. How do the different temporal resolutions of observations as well as the three applied error scenarios affect the parameter values and their uncertainty compared to the benchmark case? This issue should be discussed in light of model equifinality.

-In my opinion the authors should be more specific that their study addresses lumped hydrological models. For integrated spatially distributed models such a study surely would have different implications. Therefore I suggest to clearly state this throughout the manuscript; especially in title, introduction and discussion.

specific comments:

-In the introduction the authors provide a great overview on existing studies addressing the question how much data is needed to calibrate a hydrological model. I am wondering why the findings vary so drastically between days to years. Can the authors provide an explanation for this?

-The applications of citizen science in hydrology are broad and go beyond the collection of data. For completion the authors could mention Koch et al. (2017) were the human perception was consulted to compare the similarity between simulated spatial patterns in order to evaluate spatial performance metrics.

-Extreme outliers are filtered with respect to maximum possible streamflow values. One could imagine a more thorough filtering based on the season. An extreme outlier during low flow season can be expected to be smaller than during high flow. Have the authors considered such an improved filtering?

-I can imagine a better visualization of the data in Figure 3. Instead of nine subplots one could imagine three subplots, one for each temporal resolution. Then each error scenario could have a different color. In this way the graphs could be stretched over the entire page and the dynamics would be more visible.

[Figure]

Koch, J., & Stisen, S. (2017). Citizen science: A new perspective to advance spatial pattern evaluation in hydrology. PloS one, 12(5)

---

## Referee Comment (RC2) · Anonymous Referee #2 · 3 Aug 2018

General Comments

The manuscript titled "Value of uncertain streamflow observations for hydrological modelling" is a helpful contribution to the growing body of literature on citizen science applications in hydrology. The article is scientifically significant, is of high quality, and is well presented. The objectives of the study are clearly stated, the methods are applicable, the results are clear, and the discussion and conclusions return to the original questions posed. The overall structure of the article is sound, and the prose is for the most part acceptable. However, efforts should be made to make the language more concise by separating long sentences and properly using commas and semi colons to join dependent and independent clauses, respectively.

The following are a few more general comments. First, in the conclusion, only the

first question regarding errors and not the frequency of observations is included; it is suggested that both questions be briefly addressed. Second, the "lower benchmark" is an important part of this study, and the one sentence dedicated to it (7-13/14) doesn't provide enough information on how it was developed. Finally, additional discussion of how training could possibly decrease errors in citizen science streamflow estimates should be included (perhaps this is also included in the other paper in review). For example, should the focus be on improving depth, width, or velocity measurements? Are there any simple tools that could be added to improve the estimates? For example, could photos of the site including a person for scale (for area) and short videos (for velocity) be used to identify (and possibly filter) high error estimates?

Specific Comments (page # - line # - comment)

2-23/24 - The "stick-method" is unfamiliar and should have a reference or some description. Is this the same as the "float" method, or ?

4-7 - USBR Water Measurement Manual 2001 Ch 13.10 recommends variable surface velocity with depth

5-2 - do you have raw velocity and area data to further evaluate if the errors come more frequently from velocity or area estimates? Perhaps if you have the width and depth estimates this can also help to unpack uncertainty in areas estimates further.

6-8 - Is the one point per hour randomly selected or ??? Is hourly data a plausible citizen science output? You later say (9-21/22) that this frequency is "very unlikely." What was the frequency of the original data?

5-12 - it might be nice to more explicitly include a summary (e.g. bullet points) here of the four levels of error that you refer to later: none, low, medium, and high

7-13/14 - perhaps the range bounds on the parameters for the random selections need to be discussed further

9-25 - rather than "reduced errors" it would be better to specific either low or medium

like you do later in the sentence

9-27/28 - it would be good to consistently use either "lower benchmark" or "random parameter datasets"

10-25 - it is unclear whether "fewer data points" here is referring specifically to calibrations with only 12 observations or to calibrations with even fewer than 12 observations (which wasn't evaluated)

10-27/28 - only if the errors don't contain systematic bias; please clarify

11-7/8/9 - this sentence doesn't seem to match the main point discussed earlier in the paragraph. Earlier you state that monthly performed better than IntenseSummer and WeekendSummer which had roughly 5 times more measurements. The you say it is "easier to get a certain number of observations..." Is it rather easier to get measurements spread out through the entire year than a certain number of measurements with citizen science?

Technical Corrections (page # - line # - comment)

1-7 - "...model can be parameterized using on a limited..." need to either remove "on" or modify sentence otherwise

1-16/17 - suggest using more commas to properly phrase the content (also the last sentence of the abstract could benefit from the same)

1-29 - punctuation for the question within the sentence should be used: "... question: how much data. . . . . . are not available?"

2-14 - same issue here where you end the sentence without a question mark. Either edit similar to above or rephrase: "but the question of how informative low quality data are remains."

3-5 - should define HBV here (first use) instead of below

4-18/19 - sentence is incomplete

6-17 - it seems more logical to include Crowd52 and Crowd12 in the bullet list of the six other temporal resolutions presented

9-2 - correct grammar error "...was larger for than the..."

9-13 - which year are you referring to here: calibration or validation?

13-19 - "...this data was not statistically significant better..." needs to be revised to possible "...these data did not show statistically significant improvements in model performance..."

–––––––––––––––––––––––––––––––

---

## Author Comment (AC1) · 7 Aug 2018

**Replies to Reviewer #1**

**Anonymous Referee #1**

The manuscript entitled "Value of uncertain streamflow observations for hydrological modelling" presents interesting and novel research on the worth of citizen science discharge observations for the calibration of lumped hydrological models. The manuscript is well structured and concise with a clear motivation. The presentation and the application of the methods are scientifically sound. My comments are mostly of minor
10   character and therefore I hope to see this article soon published in HESS.

We thank the reviewer for the positive comments about our manuscript and the helpful review comments, which we address in detail below.

15  **general comments:**
-The presentation of the calibration experiments is clear and complete with regard to the model performance. However, I was wondering how the model robustness is affected by uncertain observations which was neglected by the authors. From a modelling point of view parameter uncertainty and its reduction through calibration is of high importance. Therefore I believe that an additional figure on that matter would improve the quality of the
20   study. How do the different temporal resolutions of observations as well as the three applied error scenarios affect the parameter values and their uncertainty compared to the benchmark case? This issue should be discussed in light of model equifinality.

We thank the reviewer for this helpful comment. We will include a discussion about the effects of errors in the data and the

25   effect of the timing and amount of data used for model calibration on the range of parameter values in the revised version of the manuscript.

In Figures R1-R6, we show the boxplots with the parameter ranges for each of the six catchments. Each boxplot consists of 300 values (3 year characters x 100 calibration runs). We summarized these results in another plot (Figure R7) which shows the interquartile range of the parameter distribution for each catchment for the different scenarios. The effects of the errors in

30   the data and the timing of the data used for model calibration on the interquartile range of parameter values are summarized in Table R1. The spread in the parameter values was smallest for the upper benchmark for almost all parameters and cases, although the differences were very small for some parameters (e.g. PERC, PCALT and CWH). The trend of increasing spread in the parameter range with increasing errors is clearest for the MAXBAS parameter, which is the routing parameter. The parameter range of some other parameters (e.g. TCALT, TT and BETA) also increased with increasing error in the data

35   used for calibration for some catchments, but for other parameters (e.g., CFMAX, FC, and SFCF) the temporal resolution and the number of data points used for calibration determined the range in parameter values. However, these change in the range of model parameters differed significantly for the different catchments (see differences in Figures R1-R6 and spread of the dots in Figure R7).

[Figure]

**Figure R1 Boxplots of the model parameters for different combinations of errors and temporal resolutions of the data used for model calibration for the Verzasca catchment. Each subplot shows the range for one model parameter and consists of 300 values (3 year characters x 100 calibrations). The box represents the 25th and 75th percentile, the thick horizontal line the median, the whiskers extend to 1.5 times the interquartile range below the 25th percentile and above the 75th percentile, and the dots represent the outliers. For a description of the model parameters see Table R1**Error! Reference source not found.**.**

[Figure]

**Figure R2 Boxplots of the model parameters for different combinations of errors and temporal resolutions of the data used for model calibration for the Mentue catchment. Each subplot shows the range for one model parameter and consists of 300 values (3 year characters x 100 calibrations). For a description of the box plots see Figure 1. For a description of the model parameters see Table R1.**

[Figure]

**Figure R3** Boxplots of the model parameters for different combinations of errors and temporal resolutions of the data used for model calibration for the Riale di Calneggia catchment catchment. Each subplot shows the range for one model parameter and consists of 300 values (3 year characters x 100 calibrations). For a description of the box plots see Figure 1. For a description of the model parameters see Table R1.

[Figure]

**Figure R4 Boxplots of the model parameters for different combinations of errors and temporal resolutions of the data used for model calibration for the Allenbach catchment. Each subplot shows the range for one model parameter and consists of 300 values (3 year characters x 100 calibrations). For a description of the box plots see Figure 1. For a description of the model parameters see Table R1.**

[Figure]

**Figure R5 Boxplots of the model parameters for different combinations of errors and temporal resolutions of the data used for model calibration for the Guerbe catchment. Each subplot shows the range for one model parameter and consists of 300 values (3 year characters x 100 calibrations). For a description of the box plots see Figure 1. For a description of the model parameters see Table R1.**

[Figure]

**Figure R6 Boxplots of the model parameters for different combinations of errors and temporal resolutions of the data used for model calibration for the Murg catchment. Each subplot shows the range for one model parameter and consists of 300 values (3 year characters x 100 calibrations). For a description of the box plots see Figure 1. For a description of the model parameters see Table R1.**

[Figure]

• Hourly • Weekly • Crowd52 • Monthly • Crowd12 • WeekendSpring • WeekendSummer • IntenseSummer

**Figure R7 The interquartile range of the model parameters for the six catchments for the different combinations of errors and temporal resolutions of the data used for model calibration. Each dot represents the interquartile range for one catchment (i.e. is the size of the box in Figures R1-R6). For a description of the model parameters see Table R1.**

**Table R1 Effect of errors and timing of the data used for model calibration on the interquartile ranges of the calibrated parameters in HBV-light model parameters. See also Figure R7. For a description of the different data sets (names in italic) see the main text.**

| Parameter | | Effect of errors | Effect of timing |
|---|---|---|---|
| PERC | Maximum percolation from upper to lower groundwater storage [mmd$^{-1}$] | No clear effect of errors, only *Monthly* dataset has larger range if large errors | Slightly larger range for *Monthly* and *Crowd12* data sets if large errors |
| UZL | Threshold parameter [mm] | No big effect, larger range for *Monthly* with increasing errors | Largely effect of timing |
| K0 | Storage (or recession) coefficients [h$^{-1}$] | Slightly larger parameter range for medium and large errors | No clear effect of timing |
| K1 | Storage (or recession) coefficients [h$^{-1}$] | Slightly larger range for *WeekendSpring*, *WeekendSummer IntenseSummer* datasets, smaller range for e.g. *hourly* dataset with increasing errors | *Hourly* dataset usually has the smallest range |
| K2 | Storage (or recession) coefficients [h$^{-1}$] | No effect | No effect |
| MAXBAS | Length of triangular weighting function [H] | Increasing range with increasing errors | Large range for *WeekendSpring* dataset |
| PCALT | Change in precipitation with elevation [% (100 m)$^{-1}$] | Sometimes larger and sometimes smaller range with increasing errors | No clear effect of timing, *hourly* dataset has the smallest range |
| TCALT | Change in temperature with elevation [°C (10 m)$^{-1}$] | Increasing range with increasing errors | Some effect of timing, sometimes smaller range, sometimes larger range with less data (e.g. *Weekly* dataset) |
| TT | Threshold temperature for liquid and solid precipitation [°C] | Increasing range with increasing errors | Some effect of timing |
| CFMAX | Degree-day factor [mm d$^{-1}$°C$^{-1}$] | Only for largest errors increase in parameter range | Larger range for intense summer than for others |
| SFCF | Snowfall correction factor [-] | No effect | No effect |
| CFR | refreezing coefficient [-] | No effect | No effect, *Crowd52* dataset usually has the smallest range |
| CWH | Water holding capacity of the snow storage [-] | Larger range for *WeekendSpring* and *Intense Summer* datasets with increasing errors, for other datasets no clear trend | No effect |
| FC | Maximum soil moisture storage [Mm] | *IntenseSummer* and *Weekly* datasets have larger range with increasing errors, for other datasets no clear trend | No effect |
| LP | Soil moisture value above which actual evapotranspiration reaches potential evapotranspiration [-] | No effect | No effect |
| Beta | Shape factor for the function used to calculate the distribution of rain and snow melt going to runoff and soil box | No effect | No effect |

-In my opinion the authors should be more specific that their study addresses lumped hydrological models. For integrated spatially distributed models such a study surely would have different implications. Therefore I suggest to clearly state this throughout the manuscript; especially in title, introduction and discussion.

Thank you very much for pointing this out. We agree and will adjust the text to more explicitly state that these results are for lumped hydrological models.

**specific comments:**

-In the introduction the authors provide a great overview on existing studies addressing the question how much data is needed to calibrate a hydrological model. I am wondering why the findings vary so drastically between days to years. Can the authors provide an explanation for this?

These studies all had a different focus, used different performance metrics and different definitions of what a good model performance is (see Table R2). Vrugt et al. (2006) and Yapo et al. (1996) defined stable parameters as a good calibration criterion. Others (Juston et al. (2009); Seibert and Beven (2009); Seibert and McDonnell (2015)) used benchmark calibrations and looked at the differences in the values of the objective functions. Pool et al. (2017) always used 12 streamflow values and explored the best timing of these measurements. Juston et al. (2009) used a very long time series with possibly much more variation in streamflow than is observed within one year of data (as in this study) from which the subsets were drawn. Brath et al. (2004) used a spatially distributed model and concluded that 3 months were the absolute minimum.

We will add more information to the introduction to describe why the different studies resulted in different minimum data sets and highlight even better that despite their differences they all find that limited datasets are useful.

**Table R2 Cited modelling studies focusing on the amount of streamflow data necessary to calibrate a hydrological model.**

| Study / Authors | Performance Metric | Temporal Resolution | Model | How much data was needed |
|---|---|---|---|---|
| Yapo et al., 1996 | Daily root mean square estimation criterion and hetero-scedastic maximum likelihood error | Daily | NWSRFS_SMA model (Brazil, 1988) | No more added value after 8 years of data. If wettest years are chosen for calibration, model parameters were "properly identifiable" |
| Vrugt et al., 2006 | RMSE | daily | Sacramento Soil Moisture Accounting model | Stable estimates for most of the parameters with 2-3 years of streamflow data |
| Perrin et al., 2007 | NSE in calibration, NSE and LogNSE in validation | daily | TOPMO (derived from TOPMODEL concepts (Michel et al 2003) and GR4J (Perrin et al 2003) | 350 random days out of a 39 year period including dry and wet conditions are sufficient to obtain robust model |

| | | | | parameters |
|---|---|---|---|---|
| Brath et al., 2004 | Relative volume error, relative peak error, Time to peak error | Hourly | Spatially distributed model | At least 3 months were required to obtain reliable calibration |
| Juston et al., 2009 | Combination of NSE and groundwater performance index (multi-objective calibration) | daily | HBV-Forsmark | Information content of subset of 53 days was the same, as the entire 1065-day period from which the data was drawn |
| Pool et al., 2017 | NSE and log NSE | daily | HBV | 12 data points, different "sampling" strategies (high flows, low flows, recession limbs, on the peak, etc) |
| Seibert and Beven, 2009 | NSE | daily | HBV | Model performance plateaued after 8-16 streamflow measurements within a one year period |
| Seibert and McDonnell, 2015 | the overall acceptability of a parameter set was defined by three components: (1) the model efficiency (NSE) values (Nash and Sutcliffe 1970) for the hard runoff data (calculated based on subsets of the total runoff series), (2) the acceptability of the model simulations with regard to soft data, and (3) the acceptability of the parameter values based on the experimentalist's understanding. | 10 min streamflow data | Variant of HBV | One event or 10 high flow measurements provided almost as much information as a 3 months of data |

-The applications of citizen science in hydrology are broad and go beyond the collection of data. For completion the authors could mention Koch et al. (2017) were the human perception was consulted to compare the similarity between simulated spatial patterns in order to evaluate spatial performance metrics.

Thank you for providing this reference. We will include a sentence that states that citizen science includes more than data collection in the introduction and reference this paper there.

-Extreme outliers are filtered with respect to maximum possible streamflow values. One could imagine a more thorough filtering based on the season. An extreme outlier during low flow season can be expected to be smaller than during high flow. Have the authors considered such an improved filtering?

5    We agree that when using real citizen science data more advanced filtering mechanisms are useful but these will have to be thoroughly tested first. The testing of different filtering methods is not within the scope of our study. Also, these filtering mechanisms need to be applicable for all places without measurements or local knowledge about flows. Low flows and high flows can occur in multiple seasons and differ drastically between stations (and years), since some catchments are snow influenced, others have glaciers, and others are only rain fed. Therefore, local knowledge or data about how low flows

10   depend on the season are necessary. We considered the lowest and highest ever measured values for a particular catchment size for the filtering because these data may be available for different regions and provide a very simple filter to take out the most unrealistic values. Because no extreme low flow value was replaced with the lowest ever recorded flow and only a few high flow estimates were replaced, we assume that the results would not have been significantly different if a slightly more advanced filtering mechanism for low flows was used.

15
-I can imagine a better visualization of the data in Figure 3. Instead of nine subplots one could imagine three subplots, one for each temporal resolution. Then each error scenario could have a different color. In this way the graphs could be stretched over the entire page and the dynamics would be more visible.

20   Thank you for this suggestion for an improvement. We thought about designing the graph as suggested but there are too many dots (or other symbols) that overlap as can be seen in Figures R8-R9. This makes it hard to read the figures, even if the symbols are not filled. We therefore prefer to use the nine subplots that we used in the submitted version of the manuscript. We are, however, open for other suggestions on how to improve the figure.

[Figure]

**Figure R8 Alternative design of Figure 3 with filled symbols to represent the data used for model calibration for the different scenarios.**

[Figure]

**Figure R9 Alternative design of Figure 3 with different open symbols to represent the data used for model calibration for the different scenarios.**

---

## Short Comment (SC1) · 8 Aug 2018

This is a very interesting paper which will add more value in citizen hydrology. My only comment is: When stating examples of citizen science projects that collect streamflow or stream level data (page 2 line 21), it would be helpful to include SmartPhones4Water-Nepal (http://www.smartphones4water.org/category/news/) as an example project using references of either https://link.springer.com/article/10.1007/s00267-017-0872-x or https://link.springer.com/article/10.1007%2Fs10661-018-6687-2 . This is especially important because literature about citizen science so far has been relatively focused on the west, so any example applications of citizen science in Asia should be properly included.

---

## Short Comment (SC2) · 10 Aug 2018

Thank you very much for making us aware of these interesting recent studies and the project SmartPhones4Water. We will include a reference in the revised manuscript.

---

## Author Comment (AC2) · 10 Aug 2018

General Comments

The manuscript titled "Value of uncertain streamflow observations for hydrological modelling" is a helpful contribution to the growing body of literature on citizen science applications in hydrology. The article is scientifically significant, is of high quality, and is well presented. The objectives of the study are clearly stated, the methods are applicable, the results are clear, and the discussion and conclusions return to the original questions posed. The overall structure of the article is sound, and the prose is for the most part acceptable. However, efforts should be made to make the language more concise by separating long sentences and properly using commas and semi colons to join dependent and independent clauses, respectively.

We thank the reviewer for the positive comments about our manuscript and the helpful review comments, which we address in detail below.

The following are a few more general comments.

First, in the conclusion, only the first question regarding errors and not the frequency of observations is included; it is suggested that both questions be briefly addressed.

We thank the reviewer for this helpful comment. We agree that this should be addressed as well. We will include the following sentence in the revised manuscript: **"We, furthermore, demonstrated that realistic frequencies for citizen science projects (one observation on average per week or month) can be informative for model calibration. "**

Second, the "lower benchmark" is an important part of this study, and the one sentence dedicated to it (7-13/14) doesn't provide enough information on how it was developed.

The use of upper and lower benchmarks to compare different model results follows the strategy of several recent studies (van Meerveld et al., 2017; Pool et al., 2017; Wang et al., 2017). Seibert et al. (2018) point out that it is important to assess what model performance is possible (upper benchmark) because the data used for model calibration and validation contain errors and a perfect model fit can't be expected, and to compare the model performance to what can be expected (lower benchmark) because the driving (precipitation and temperature) data often dictate that models can't be too far off for humid catchments, as long as the water balance is respected. The lower benchmark used in this study is therefore the median model performance for an uncalibrated model (based on 1000 random parameter sets).
We will extend the section in the revised manuscript:
**"In humid climates, the input data (precipitation and temperature) often dictate that model simulations can't be too far off as long as the water balance is respected (Seibert et al., 2018). To assess the value of limited inaccurate streamflow data compared to a situation without any streamflow data, a lower benchmark (Seibert et al., 2018) was therefore used as well. Here the lower benchmark was defined as the median performance of the model ran with 1000 random parameters sets. By running the model with 1000 randomly chosen parameter sets, we represent a situation where no streamflow data for calibration are available and the model is driven only by the temperature and precipitation data. We used 1000 different parameter sets to cover most of the model variability due to the different parameter combinations."**

References:
van Meerveld, H J; Vis, Marc J P; Seibert, Jan (2017). Information content of stream level class data for hydrological model calibration. Hydrology and Earth System Sciences, 21(9):4895-4905.

5 Pool, Sandra; Vis, Marc J P; Knight, Rodney R; Seibert, Jan (2017). Streamflow characteristics from modeled runoff time series – importance of calibration criteria selection. Hydrology and Earth System Sciences, 21(11):5443-5457.

Seibert, Jan; Vis, Marc J P; Lewis, Elizabeth; van Meerveld, H J (2018). Upper and lower benchmarks in hydrological modelling. Hydrological Processes, 32(8):1120-1125.
10

Wang, Ling; van Meerveld, H J; Seibert, Jan (2018). Effect of observation errors on the timing of the most informative isotope samples for event-based model calibration. Hydrology, 5(1):4.

Finally, additional discussion of how training could possibly decrease errors in citizen science stream-
15 flow estimates should be included (perhaps this is also included in the other paper in review). For example, should the focus be on improving depth, width, or velocity measurements? Are there any simple tools that could be added to improve the estimates? For example, could photos of the site including a person for scale (for area) and short videos (for velocity) be used to identify (and possibly filter) high error estimates?
20

There are indeed multiple possibilities for training. These include tutorial videos, or providing a list with well-known streams and their ranges in width, depth, flow velocity and streamflow to indicate ball park numbers.
We will include a brief statement on potential training options. However we do not want to focus too much on potential training options because their advantages and effectiveness are not known yet: **"Options for training might be tutorial**
25 **videos, as well as providing values for the width, average depth and flow velocity of some well-known streams (Strobl et al., in review)."**

**Specific Comments (page # - line # - comment)**
2-23/24 - The "stick-method" is unfamiliar and should have a reference or some description.
30 Is this the same as the "float" method, or ?

We will clarify this by rewriting these sentences in the following way: "**Estimating streamflow is obviously more challenging than reading levels from a staff gauge but citizens can apply the stick or float method, where they measure the time it takes for a floating object (e.g., a small stick) to travel a given distance to estimate the flow velocity. Com-**
35 **bined with estimates for the width and the average depth of the stream, this allows them to obtain a rough estimate of the streamflow**."

4-7 - USBR Water Measurement Manual 2001 Ch 13.10 recommends variable surface velocity with depth
40

We are unfortunately not exactly sure what this comment refers to. We used a factor of 0.8 to correct for the decline in flow velocity with depth and to obtain an average velocity from the surface velocity. Text books (e.g. Harrelson, Rawlins, & Potyondy, 1994) recommend this correction factor. Hauet et al. (in review) and Morlot et al. (2018) showed that this correction factor is reasonable for most streams, except for concrete channels (see page 5, line 7 in the first submitted version). Even if
45 the exact value of the correction factor is uncertain (e.g. varies between 0.6 and 0.95), the impact on the estimated streamflow is small compared to the errors in the estimates of the velocity, width and depth.

References:

Harrelson, C.C., Rawlins, C.L. & Potyondy, J.P., 1994. Stream channel reference sites: an illustrated guide to field technique. (http://www.treesearch.fs.fed.us/pubs/20753)

Hauet, A., Morlot, T. & Daubagnan L. (in review) Velocity profile and depth-averaged to surface velocity in natural streams: a review over a large sample of rivers.

Morlot T., Hauet, A., & L. Daubagnan, L., 2018. Computation of the coefficient relating depth-averaged velocities to surface velocity over a large sample of French cross-sections gauged with a current meter, Geophysical Research Abstracts, Vol. 20, EGU2018-1874.

5-2 - do you have raw velocity and area data to further evaluate if the errors come more frequently from velocity or area estimates? Perhaps if you have the width and depth estimates this can also help to unpack uncertainty in areas estimates further.

Strobl et al. (in review) show that the width can generally be estimated better than the depth and velocity. Here we would like to focus on the value of the resulting streamflow estimates for hydrological modelling. We will mention that the depth is particularly uncertain when we describe the options of training.

6-8 - Is the one point per hour randomly selected or ??? Is hourly data a plausible citizen science output? You later say (9-21/22) that this frequency is "very unlikely." What was the frequency of the original data?

The measurements from the Swiss Federal Office for the Environment (FOEN) have a 10 minute interval. The values we used were hourly averages. Hourly data were used to run the model because this is the resolution of the precipitation data, and represents the highest resolution that is regularly used for hydrological models in Switzerland and the HBV-model. We used the hourly data also for the simulations with error, even though it is very unlikely to get such a high contribution rate for citizen science projects, because this allowed us to draw conclusions about the effects of errors (i.e. for cases where the temporal resolution is "optimal" and only the quality is bad).

We will insert the following sub-sentence in the manuscript: **"Hourly runoff time series (based on 10 minute measurements) for the six study catchments were obtained from the Federal Office for the Environment (FOEN; see Table 1 for the gauging station numbers)."**

and the following text after we describe the different scenarios:

**"Except for the hourly data, these scenarios were based on our own experiences within the CrowdWater project (www.crowdwater.ch) and information from the CrowdHydrology project (Lowry and Fienen, 2013). The hourly dataset was included to be able to test the effect of the errors when the temporal resolution of the data is optimal (i.e., by comparing simulations with the hourly FOEN data and those with hourly data with errors)."**

5-12 - it might be nice to more explicitly include a summary (e.g. bullet points) here of the four levels of error that you refer to later: none, low, medium, and high

In the revised paper we will include the following list:

**"To summarize, we tested the following four cases:**

- **No error: The data measured by the FOEN, assumed to be error-free, the benchmark in terms of quality.**
- **Small error: random errors according to the log-normal distribution of the snapshot campaigns with the standard deviation divided by 4.**
- **Medium error: random errors according to the log-normal of the snapshot campaigns with the standard deviation divided by 2.**
- **Large error: typical errors of citizen scientists, i.e. random errors according to the log normal distribution of errors from the snapshot campaigns."**

7-13/14 - perhaps the range bounds on the parameters for the random selections need to be discussed further

We agree that Table 1 in the supplemental material with the range of the parameters should be mentioned in the text. We will add a sentence in chapter 2.6: **"The parameters were calibrated within the typical ranges of the parameters (see Supplemental Material – Table 1)."**

9-25 - rather than "reduced errors" it would be better to specific either low or medium like you do later in the sentence

We agree. We will change the sentence into: **"With medium errors, however, and one data point per week on average or regularly spaced Monthly data, the data were informative for model parameterization."**

9-27/28 - it would be good to consistently use either "lower benchmark" or "random parameter datasets"

We agree. We will change it to **"lower benchmark"**.

10-25 - it is unclear whether "fewer data points" here is referring specifically to calibrations with only 12 observations or to calibrations with even fewer than 12 observations (which wasn't evaluated)

We agree that this statement is unclear. With "Fewer data points" we meant that the performance of models generally decreased faster with increasing errors if 12 instead of 48-52 data points were available. We will rewrite this sentence:
**"…the results of this study also suggest that the performance of models decreases faster with increasing errors when fewer data points are available (i.e. there was a faster decline in model performance with increasing errors for models calibrated with 12 data points than for the models calibrated with 48-52 data points)."**

10-27/28 - only if the errors don't contain systematic bias; please clarify

Indeed, errors only average out when more data points are included if the errors don't contain a systematic bias. Our errors include a small overestimation but apparently the effect of this small bias is not strong. We will change the sentence:
**"These findings can be explained by the compensating effect of the number of observations and their accuracy because the random errors for the inaccurate data average out when a large number of observations are used, as long as the data do not have a large bias."**

11-7/8/9 - this sentence doesn't seem to match the main point discussed earlier in the paragraph. Earlier you state that monthly performed better than IntenseSummer and WeekendSummer which had roughly 5 times more measurements. The you say it is "easier to get a certain number of observations…" Is it rather easier to get measurements spread out through the entire year than a certain number of measurements with citizen science?

Thanks for pointing at these confusing statements
The statement at 11-7/8/9 points to the fact that it is likely easier to obtain a certain number of observations distributed over the year than at very specific times or flow conditions because people can contribute whenever they want . The goal of the statement that the Monthly dataset performs better than the IntenseSummer and WeekendSummer datasets is to make it clear that fewer data can be more useful if they are distributed over the entire year (likely because they contain more information on the streamflow-variability). The term "certain number" is therefore confusing and not necessary. We will delete it and rewrite the sentence: **"This is good news for using citizen science data for model calibration as it suggests that the tim-**

**ing is not as important as the number of observations because it is likely much easier to get observations throughout the year than during specific periods or flow conditions."**

**Technical Corrections (page # - line # - comment)**

5     1-7 - "….model can be parameterized using on a limited…" need to either remove "on" or modify sentence otherwise

Thanks for this suggestion for improvement. We will delete the word "on".

10     1-16/17 - suggest using more commas to properly phrase the content (also the last sentence of the abstract could benefit from the same)

Thanks for making us aware of this, we changed the indicated sentence accordingly: "**These included scenarios with one observation each week or month, as well as scenarios that are more realistic for crowdsourced data that generally**

15     **have an irregular distribution of data points throughout the year, or focus on a particular season**."

1-29 - punctuation for the question within the sentence should be used: …question: how much data… are not available?"

20     Thanks for this suggestion. We will change it accordingly.

2-14 - same issue here where you end the sentence without a question mark. Either edit similar to above or rephrase: "but the question of how informative low quality data are remains."

25     Thanks for pointing out also the second case. We will change it to: "**These results are encouraging for the calibration of hydrological models for ungauged basins based on a limited number of high quality measurements, but the question remains: how informative are low quality data?**"

3-5 - should define HBV here (first use) instead of below

30

Thanks for pointing this out, we will change it accordingly.

4-18/19 - sentence is incomplete

35     Thanks for pointing this out. We changed it to: "**For the validation, we chose the year closest to the mean summer streamflow and the years with the lowest and the highest summer streamflow sums (see Table 2).**"

6-17 - it seems more logical to include Crowd52 and Crowd12 in the bullet list of the six other temporal resolutions presented

40

We agree: We will adapt the section in the revised manuscript.

9-2 - correct grammar error "…was larger for than the…"

45     Thank you for pointing this out, we will correct it.

9-13 - which year are you referring to here: calibration or validation?

Thanks for pointing at this shortcoming: We will edit the sentence to clarify this: **"For 13 out of the 18 catchment and year combinations, the Crowd52 datasets with fewer than 10 % high streamflow data points led to a better validation performance than the Crowd52 datasets with more high streamflow data points."**

13-19 - "…this data was not statistically significant better…" needs to be revised to possible "…these data did not show statistically significant improvements in model performance…"

Thanks for this suggestion for improvement. We will change the sentence in the revised version into: **"… (i.e. the median performance of the models calibrated with these data was not significantly better than the median performance of the models with random parameter values)."**

---

## Author Response (AR1)

**Replies to Editor**

Editor Decision: Publish subject to minor revisions (further review by editor) (10 Sep 2018) by Nadav Peleg

5 Comments to the Author: Dear Simon Etter and co-authors,

Thank you for posting your responses to the two referees' reports. The reviews are quite favorable, but they also raised some important comments and suggestions that I urge you to consider as they might improve the quality of the manuscript. Based on my own reading, I find this to be an interesting paper that fits the scope of HESS

well and will be of interest to the community.

We thank the editor and reviewers for the positive feedback on our manuscript. Please see the answers to the individual comments of the editor and reviewers below.

15

10

In addition to the comments from the reviewers, I kindly ask you to also consider my two cents: (i) consider discussing how general are the presented results (e.g. considering different climates, terrain, hydrological model types and calibration techniques);

- 20 We agree that it is of interest to all readers to what extent these results can be generalized. We added a section at the beginning of the discussion in section 4.1 (page 10, lines 10-17): "In this study, we evaluated the information content of streamflow estimates by citizen scientists for calibration of the bucket-type hydrological model for six Swiss catchments. Streamflow estimates by citizens are sometimes very different from the measured values, and the individual estimates can be dis-informative for model calibration (Beven, 2016; Beven and Westerberg, 2011). While the hydro-
- 25 climatic conditions, the model or the calibration approaches might be different in other studies, these results should be applicable for a wide range of cases. However, for physically-based spatially distributed models that are usually not calibrated automatically, the use of limited streamflow data would probably benefit from a different calibration approach. Furthermore, our results might not be applicable in arid catchment cases where rivers fall dry for some period of the year because the linear reservoirs used in the HBV model are not appropriate for such systems."
- 30

(ii) Fig. 3 – consider reducing the symbol size of the dots and zooming closer to the observed hydrograph, I found it difficult to see the match between obs. and sim. data;

We thank the editor for this helpful suggestion. We changed the figure in the manuscript. Because not all dots are within the

35 figure limits anymore, we also added the number of outliers and a sentence to the figure caption to explain this.

(iii) Fig. 6 – consider focusing on one or two cases (e.g. calibration: dry and validation: dry and calibration: wet and validation: wet) to make the figure larger and clearer, and presenting the other cases as supplementary information.

We changed the figure 6 to contain only the two suggested cases and added the entire figure to the supplemental material. The plot in the supplementary material also shows the results for the two cases highlighted in Figure 6 to enable an easier comparison between all cases.

5 I invite you to upload a revised manuscript, incorporating the proposed changes and additions, and making any other modifications where you see fit (minor revision iteration). I look forward to receiving the revised manuscript.

Sincerely, Nadav Peleg

**Replies to Reviewer #1**

**Anonymous Referee #1**

Received and published: 13 July 2018

5

10

The manuscript entitled "Value of uncertain streamflow observations for hydrological modelling" presents interesting and novel research on the worth of citizen science discharge observations for the calibration of lumped hydrological models. The manuscript is well structured and concise with a clear motivation. The presentation and the application of the methods are scientifically sound. My comments are mostly of minor character and therefore I hope to see this article soon published in HESS.

We thank the reviewer for the positive comments about our manuscript and the helpful review comments, which we address in detail below.

**15 general comments:**

-The presentation of the calibration experiments is clear and complete with regard to the model performance. However, I was wondering how the model robustness is affected by uncertain observations which was neglected by the authors. From a modelling point of view parameter uncertainty and its reduction through calibration is of high importance. Therefore I believe that an additional figure on that matter would improve the quality of the

20 study. How do the different temporal resolutions of observations as well as the three applied error scenarios affect the parameter values and their uncertainty compared to the benchmark case? This issue should be discussed in light of model equifinality.

We thank the reviewer for this helpful comment. We included a short paragraph about the effects of the errors on the timing

- 25 and amount of data used for model calibration on the range of parameter values in the revised version of the manuscript. In Figures R1-R6, we show the boxplots with the parameter ranges for each of the six catchments. Each boxplot consists of 300 values (3 year characters x 100 calibration runs). We summarized these results in another plot (Figure R7) that shows the interquartile range of the parameter distribution for each catchment for the different scenarios. The effects of the errors in the data and the timing of the data used for model calibration on the interquartile range of parameter values are summarized
- 30 in Table R1. The spread in the parameter values was smallest for the upper benchmark for almost all parameters and cases, although the differences were very small for some parameters (e.g. PERC, PCALT and CWH). The trend of increasing spread in the parameter range with increasing errors is clearest for the MAXBAS parameter, which is the routing parameter. The parameter range of some other parameters (e.g. TCALT, TT and BETA) also increased with increasing error in the data used for calibration, but for other parameters (e.g., CFMAX, FC, and SFCF) the temporal resolution and the number of data
- 35 points used for calibration determined the range in the optimized parameter values. However, these changes in the range of model parameters differed significantly for the different catchments (see differences in Figures R1-R6 and spread of the dots in Figure R7). Because these trends are not very clear, we prefer not to include any of the Figures R1-R6 but added a section 3.5 in the results on the parameter distribution (section 3.4, page 9, lines 14-19): "For most parameters the spread in the optimized parameter values was smallest for the upper benchmark. The spread in the parameter values increased

with increasing errors in the data used for calibration, particularly for MAXBAS (the routing parameter) but also for some other parameters (e.g. TCALT, TT and BETA). However, for some parameters (e.g., CFMAX, FC, and SFCF) the range in the optimized parameter values was mainly affected by the temporal resolution of the data and the number of data points used for calibration. It should be noted though that the changes in the range of model parameters differed significantly for the different catchments and the trends weren't very clear."